# Bisulfite treatment and single-molecule real-time sequencing reveal D-loop length, position, and distribution

**Shanaya Shital Shah[1], Stella R Hartono[2], Frédéric Chédin[2], Wolf-Dietrich Heyer[1,2]***

[1]Department of Microbiology and Molecular Genetics, University of California, Davis, Davis, United States; [2]Department of Molecular and Cellular Biology, University of California, Davis, Davis, United States

**Abstract** Displacement loops (D-loops) are signature intermediates formed during homologous recombination. Numerous factors regulate D-loop formation and disruption, thereby influencing crucial aspects of DNA repair, including donor choice and the possibility of crossover outcome. While D-loop detection methods exist, it is currently unfeasible to assess the relationship between D-loop editors and D-loop characteristics such as length and position. Here, we developed a novel in vitro assay to characterize the length and position of individual D-loops with near base-pair resolution and deep coverage, while also revealing their distribution in a population. Non-denaturing bisulfite treatment modifies the cytosines on the displaced strand of the D-loop to uracil, leaving a permanent signature for the displaced strand. Subsequent single-molecule real-time sequencing uncovers the cytosine conversion patch as a D-loop footprint. The D-loop Mapping Assay is widely applicable with different substrates and donor types and can be used to study factors that influence D-loop properties.

## Introduction

Homologous recombination (HR) is a universal DNA repair pathway that is vital to genome maintenance and the repair of double-stranded DNA breaks, stalled or collapsed forks and inter-strand crosslinks (*Kowalczykowski et al., 2016*; *San Filippo et al., 2008*; *Wright et al., 2018*). Repair by HR begins by resection of the broken DNA molecule leaving a single-stranded DNA (ssDNA) with a free 3′-OH end. The Rad51 protein forms a filament on the ssDNA and carries out DNA strand invasion into a homologous duplex donor. The Rad54 motor protein translocates along the three-strand intermediate that has been formed by the invasion (*Wright and Heyer, 2014*). It simultaneously displaces Rad51, while threading out a heteroduplex DNA (hDNA) and a displaced strand. This activity results in the formation of a metastable intermediate called a displacement loop (D-loop). The D-loop comprises a single-stranded displaced strand, an hDNA, and a DNA strand-exchange junction at each extremity of the hDNA. The D-loop can then be extended by DNA synthesis, using the donor DNA as a template. In a pathway known as synthesis-dependent strand annealing (SDSA), the extended D-loops are disrupted, allowing the newly extended end to anneal to the second end of the break. SDSA always results in a non-crossover (NCO) outcome. Alternatively, the second end of the break can anneal to the displaced strand of the extended D-loop, leading to the formation of a double-Holliday junction (dHJ) (*Kowalczykowski, 2015*; *Pâques and Haber, 1999*). Theoretically, a dHJ may also form when both ends of the broken DNA simultaneously invade a donor, as discussed in *Wright et al., 2018*. The dHJ may then be dissolved into a non-crossover or nucleolytically resolved into a non-crossover or a crossover outcome (*Wright et al., 2018*). Crossover products are disfavored in somatic cells as they have the potential to lead to genomic rearrangements or loss of heterozygosity (*Li and Heyer, 2008*).

*For correspondence:
wdHeyer@ucdavis.edu

## D-loops are key intermediates in HR

The D-loop intermediate is a vital regulatory point in HR, and several factors regulate D-loop formation as well as disruption (*Wright et al., 2018*). Factors that form a D-loop may affect D-loop properties such as D-loop position and length, which may, in turn, influence the likelihood of a D-loop disruption. The following D-loop properties influence D-loop disassembly. First, the position of a D-loop will influence whether or not the 3′-OH end of invading DNA is incorporated into the hDNA. D-loops containing an annealed 3′-OH end are primed for an extension by a DNA polymerase and are less likely to be reversed (*Li and Heyer, 2009*). Conversely, D-loops with a 3′-flap may become a loading pad for D-loop disruption enzymes. Helicases and/or topoisomerases such as Sgs1-Top3-Rmi1, Mph1, and Srs2 reverse D-loops with different specificities (*Fasching et al., 2015*; *Liu et al., 2017*; *Piazza et al., 2019*; *Prakash et al., 2009*; *Putnam et al., 2009*). Internal D-loops also increase the possibility of a second invasion near the 3′-end, either in the same donor (*Wright et al., 2018*) or a different donor DNA (multi-invasions) and the formation of potential genomic rearrangements (*Piazza and Heyer, 2018*). Second, the presence of mismatches within hDNA enhances the D-loop disruption in a process termed heteroduplex rejection (*Chakraborty et al., 2016*; *Honda et al., 2014*). Thus, D-loops formed ectopically at heterologous sites would have a higher likelihood of reversal than at fully homologous donors, influencing the choice of the donor (*Chakraborty et al., 2016*). Third, the length of a D-loop might influence D-loop stability and likely their susceptibility to disruption enzymes. Long extended D-loops are also more likely to have second-end capture, leading to dHJ formation and the possibility of a crossover outcome (*Wright et al., 2018*). In summary, D-loop properties that influence D-loop reversibility regulate donor choice, the possibility of forming dHJ, and a crossover outcome, as well as multi-invasions and resultant genomic rearrangements (*Piazza and Heyer, 2018*; *Piazza et al., 2019*). Thus, these features underline the importance of studying D-loop characteristics.

Furthermore, several interlinked factors influence D-loop characteristics. First is the resection length, where longer resected tracts permit the formation of longer D-loops. However, in vivo hypo-resection is usually rare, as evidenced by the highly processive resection machinery (*Mimitou and Symington, 2011*). As resection tract length increases, it allows for more possible internal invasions, influencing D-loop position, length, and potentially also donor choice. Internal D-loops could be disrupted to try end-invasion again, the hDNA may be extended or migrated towards the 3′-end, or the 3′-flap may be cleaved or involved in a second invasion at a different site (*Wright et al., 2018*). Second, Rad51 filament properties such as length, position, composition (inclusion of accessory factors) and stability, may influence Rad54 translocation activity with subsequent effects on D-loop characteristics. Third, the length of homology between the invading and donor DNA decides the longest possible D-loop length. Subsequently, long homology lengths are associated with increased crossovers (*Inbar et al., 2000*). Fourth, the topology at donor site or chromatin remodelers altering DNA topology may influence the position and length of a D-loop that is highly favored in negatively supercoiled regions (*Wright et al., 2018*). Lastly, the extent of DNA synthesis may or may not directly reflect the length of the extended D-loop. The nascent D-loop can either be extended past the initial point of invasion with DNA synthesis or be migrated along with DNA synthesis. Thus, the gene conversion tracts (*Guo et al., 2017*; *Neuwirth et al., 2007*) that reflect the length of recombination-associated DNA synthesis do not necessarily mirror the length of the D-loop. Yet gene conversion tract lengths positively correlate with a crossover outcome (*Guo et al., 2017*). In summary, D-loop properties, as well as factors that influence D-loop properties, impact the repair outcome. Thus, it is crucial that an assay that permits the characterization of D-loops at the single-molecule level to be developed.

## Current methods for D-Loop detection and their limitations

Since D-loops are dynamic in nature, it has been challenging to develop an experimental method to study D-loops. Recent advancements in the study of nascent and extended D-loop dynamics includes the development of the D-loop capture and the D-loop extension assays, as well as modifications to a widely-used assay called the D-loop assay (*Piazza and Heyer, 2018*; *Piazza et al., 2019*; *Wright and Heyer, 2014*). Though D-loop assays typically utilize a short,~100 nt ssDNA as the broken strand, *Wright and Heyer, 2014* showed that D-loops formed using physiological length ssDNA substrates with a 5′-duplex DNA may better recapitulate D-loop dynamics in vitro.

*Wright and Heyer, 2014* also developed a restriction-digestion based assay to map the location of D-loops across the region of homology. However, the assay is limited by the presence and distribution of unique restriction sites across the homologous region. Moreover, the method relies on the use of a supercoiled duplex donor to restrict D-loop length. Additionally, the distance between two restriction sites should be greater than the maximum possible D-loop length in a supercoiled donor to distinguish between the D-loops formed at the two sites. Hence, while the restriction-digestion mapping assay broadly reveals D-loop distribution in a population of in vitro D-loops, it cannot define the length of individual D-loops.

In vivo detection of D-loops has been even more challenging, especially somatic D-loops. Meiotic single-end invasions (SEIs) that likely represent a particular form of extended D-loop destined to becoming a crossover are detected using 2D-gel electrophoresis (*Hunter and Kleckner, 2001*). Recently, using a proximity-ligation-based method, Piazza and colleagues demonstrated the ability to quantitatively measure somatic nascent D-loop formation in vivo (*Piazza et al., 2018*). The sensitivity of the assay relies on preserving the D-loops through psoralen-crosslinking. However, the crosslinking efficiency limits the assay by prohibiting the detection of D-loops shorter than the crosslinking density of ~500 bp. Thus, the assay falls short in the ability to distinguish a change in D-loop signal caused by a change in D-loop length or an alteration in D-loop quantity among a population of cells. Other assays either detect extended D-loops, a downstream product of nascent D-loop (*Hicks et al., 2011*; *Piazza et al., 2018*), or measure the extent of DNA synthesis *via* gene conversion tracts (*Guo et al., 2017*; *Neuwirth et al., 2007*), and thus, do not reflect the dynamic nascent D-loop properties. Thus, there is an unmet need for a method allowing the study of D-loop properties, such as its length, position, and distribution.

Here, we developed a novel high-throughput assay to detect for the first time the length and position of D-loops formed in vitro at the single-molecule level and near base-pair resolution. Subsequent analysis of individual D-loops reveals the distribution of D-loop lengths and position in a population of D-loops. Lack of enrichment or a crosslinking step permits relatively unbiased D-loop analysis. The assay is based on non-denaturing bisulfite sequencing, adapted from the approach used to map R-loops that are formed by RNA invasion in DNA (*Malig et al., 2020*; *Yu et al., 2003*). The method allows easy multiplexing of several D-loop samples with high coverage. The method is accompanied by computational analysis and data visualization pipeline for D-loop profiling. The method is widely applicable to study (i) D-loops formed with a wide range of ssDNA substrates and donor DNA, (ii) factors altering D-loop properties (iii) D-loop disruption factors and their preferences for D-loop features. We also discuss the limitations and possible refinements of this method.

## Results

### Setting up the D-loop mapping assay with several layers of internal control

To establish and optimize the D-loop Mapping Assay (DMA), we started by using a size-restricted D-loop sample formed in vitro. To do this, we used a supercoiled plasmid as a dsDNA donor to perform an in vitro D-loop reaction (*Figure 1A*; *Wright and Heyer, 2014*). For every 10.4 bp of hDNA generated in the D-loop, one negative supercoil from the dsDNA is consumed. Plasmids purified from *E. coli* have a mean supercoiling density of σ = −0.07 supercoil/turn (*Champion and Higgins, 2007*), which we verified independently (*Solinger et al., 2002*). Thus three kbp plasmids such as those used in this study should contain ~20 ± 5 negative supercoils, meaning that they can accommodate a maximum of ~210 ± 50 bp hDNA, even if the length of the invading homologous ssDNA is longer. Longer D-loops would cause the introduction of compensatory positive supercoils, which is energetically unfavorable. By contrast, fully relaxed plasmids carrying D-loops ~ 210 bp long are expected to be in the lowest possible energy state and thus favored at equilibrium (*Wright et al., 2018*). Thus, such an experimental setup allowed us to establish the DMA by predefining the expected lengths of the D-loops.

Once formed, the D-loop reactions were stopped and subjected to bisulfite treatment at room temperature to allow cytosine-to-uracil conversions on the single-stranded displaced strand. Using primers that are non-biased to conversion and outside the region of homology, the DNA was PCR amplified, purified, and subjected to single-molecule real-time sequencing (*Figure 1B*). The

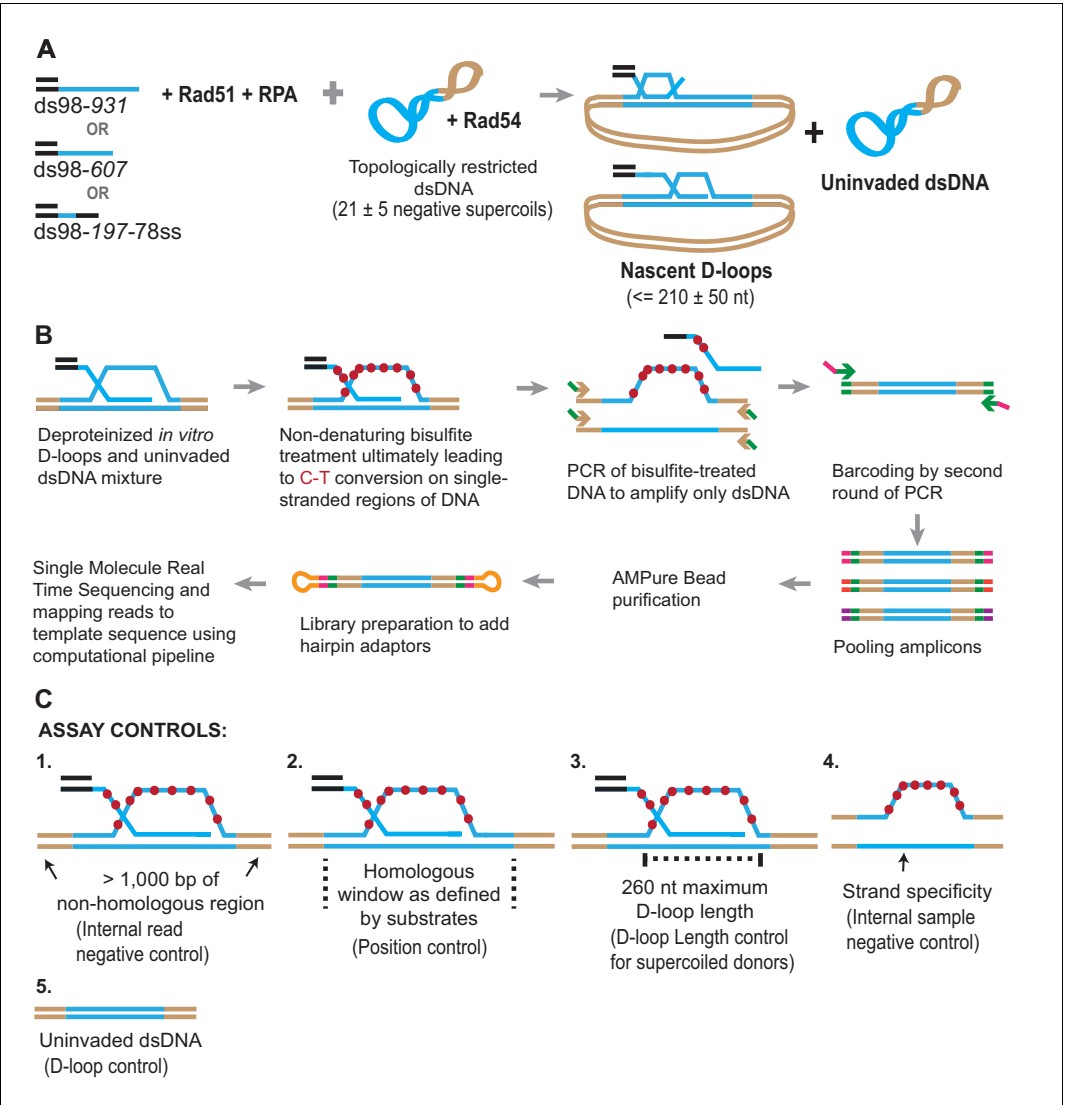

**Figure 1.** Schematic of a novel D-loop Mapping Assay (DMA). (**A**) Schematic of an in vitro D-loop reaction with a supercoiled dsDNA donor. Substrates have different lengths of homology (shown in blue line) to the supercoiled dsDNA donor. The substrates also have a 98 bp 5′ non-homologous duplex DNA to mimic physiological invading DNA. The D-loops would be restricted to be ~210 ± 50 nt in size, due to the supercoiling density of the donor DNA. (**B**) Schematic of the D-loop mapping assay (DMA) (for details, see Materials and methods). For a D-loop on a supercoiled dsDNA, the dsDNA is cropped here to only depict the strand invasion. Primers specific to the donor outside the region of homology (indicated by brown arrowheads) and having an additional universal primer sequence (in green) were used. Initially, only the primer sequence indicated by brown arrowheads would anneal to the target DNA. Barcodes were added to the amplicons *via* a second round of PCR using the universal primer sequence (green arrow ahead indicates primer location). Hairpin adaptors were added during library preparation (shown in orange), for single-molecule real-time sequencing. (**C**) Schematic depiction of the five internal controls for the DMA.

sequencing reads were then processed, mapped to the reference sequence, and analyzed to reveal cytosine conversion regions as D-loop footprints. The cytosine conversion regions were defined as D-loop footprints only when they crossed a pre-specified peak threshold. The peak threshold was implemented to filter out any background cytosine conversions from spontaneous DNA breathing (for details, see Materials and methods).

Negative controls at each step of the analysis further enhanced the strength of this well-defined in vitro design (*Figure 1C*). First, a > 1,000 bp non-homologous dsDNA region, where no D-loop

footprint is expected, flanked the homologous region. This serves as an internal negative control within each sequencing read. Second, the homology window itself can be further altered by using ssDNA substrates of different lengths, varying in this study from 197 nt to 931 nt. Third, D-loop footprints should not exceed 260 nt for D-loops formed using supercoiled dsDNA donors. This control holds true only for topologically restricted supercoiled donors, and not linear dsDNA donors. Fourth, we expect D-loop footprints to exist only on reads originating from the top strand of dsDNA donor plasmids that is displaced upon strand invasion. The bottom-strand, by contrast, should not harbor any footprints. This represents a further internal control within each sample. Lastly, an experimental negative control can be generated by using a D-loop reaction performed in the absence of invading ssDNA or the Rad51 recombinase. Without ssDNA or Rad51, no strand invasion footprint is expected. Thus, these predicted positions and/or length of D-loops will provide confidence that the footprints derived solely from D-loops.

## D-loops are positioned within the homology window, while D-loop length is restricted by the supercoiling density of the dsDNA donor

### D-loop footprints are strand-specific, and D-loop levels detected by DMA correlate with gel-based quantitation

We implemented the controlled experimental set-up described above and analyzed in vitro D-loops formed using three ssDNA substrates with different homology lengths of 197 nt (ds98-*197*-78ss), 607 nt (ds98-*607*) and 931 nt (ds98-*931*). Before subjecting to DMA, D-loop formation with each substrate was confirmed by gel visualization (*Figure 2—figure supplement 1A–C*). The results from DMA obtained using each substrate type and a supercoiled donor are shown as footprint maps in *Figure 2A–C*. For the D-loops with ds98-*931*, 264 individual D-loop footprint-containing reads were recovered out of a total of 1,340 reads from the top strand (*Figure 2A*). Overall, we detected 485 independent D-loop footprints among 3,621 top-strand reads over >3 replicates, representing a 13.4% formation efficiency (*Figure 2D*). These D-loop footprints were overwhelmingly strand-specific, as expected. Only 26 (0.43%) footprint-containing reads were recovered out of nearly 6,000 bottom-strand reads (*Figure 2D*). Thus, the top and the bottom-strand show a > 30 fold difference in their bisulfite reactivity. Similarly, for the D-loops with ds98-*607* substrate, 310 (11.59%) of the 2,673 top-strand reads had a D-loop footprint. While only 6 (0.1%) of the 5,797 bottom-strand reads had a footprint (>100 fold lower than the percentage of footprints found on the top-strand) (*Figure 2D*). Lastly, with the ds98-*197*-78ss substrate, 75 (6.1%) of the 1,229 top-strand reads and 7 (0.2%) of the 3,494 bottom-strand reads had a footprint. Importantly, when D-loop reactions were performed in the absence of Rad51 or ssDNA, no footprints were observed among 3,119 top-strand reads and 2,651 bottom-strand reads. Thus, the D-loop footprints recovered by DMA depict the expected strand-specificity and DNA strand invasion-specificity.

Note that for the D-loop samples, usually, there was a > 2 fold abundance of the bottom-strand reads compared to the top-strand reads. This is because during the bisulfite modification of cytosines, there is some unavoidable nicking of the ssDNA. Since with the D-loop samples, only the top strand is expected to be bisulfite-modified, nicking leads to unavoidable loss of some top-strand DNA post PCR amplification. Confirming this, the samples devoid of a D-loop, such as those without Rad51 or ssDNA, had a comparable number of top and bottom-strand reads. Thus, the difference in the number of total top and bottom-strand reads we detected only reflects the specificity of the assay for imprinting D-loops.

In line with this loss of some top-strand molecules, the percentage of top-strand reads containing a D-loop footprint (% D-loops from the DMA) was always lower than the percentage of D-loops visualized on a gel (*Figure 2E*). The D-loop levels from both gel-based (*Figure 2—figure supplement 1A–C*) and DMA assay (*Figure 2D*) were quantified relative to the uninvaded donor. With the ds98-*931* substrate, 22% D-loops were quantified from the gel-based detection, while ~13% D-loops were measured from the DMA assay (*Figure 2E*). Similar observations were made for the substrates with shorter homologies: ds98-607, 21% gel assay and 12% DMA; ds98-*197*-78ss, 12% gel assay and 7% DMA. Thus, for all three substrates, consistently,~60% of the D-loops visualized on the gel were detected by the DMA assay. While *Figure 2D* depicts the total reads analyzed for each substrate, data from each replicate are available in *Source data 1*. In summary, despite slightly lower D-loop

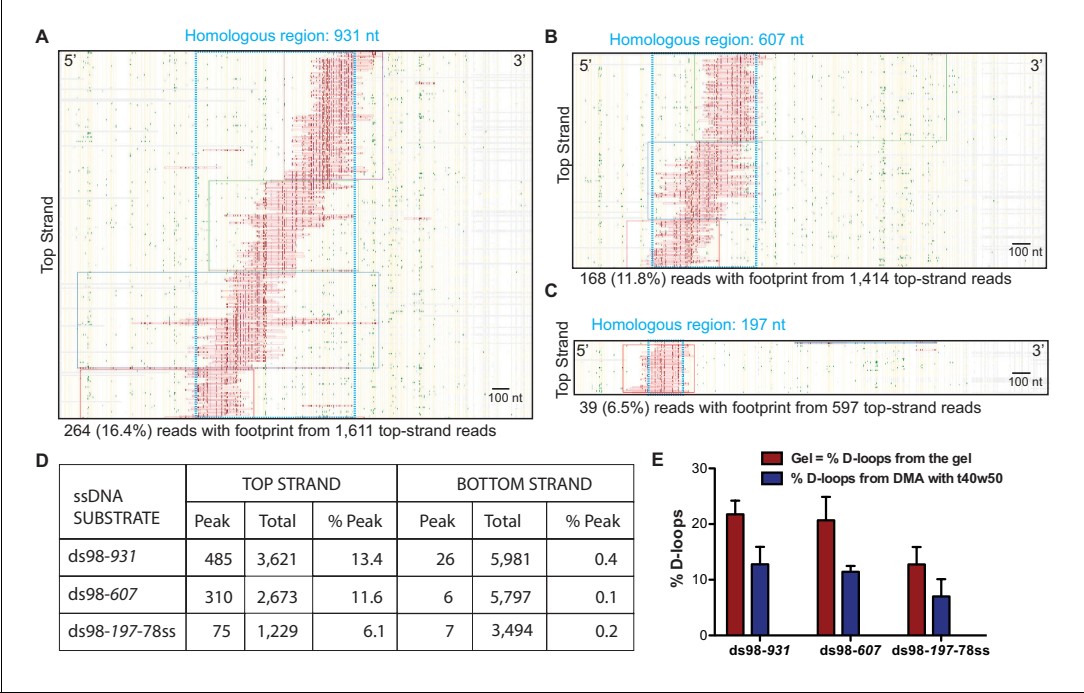

**Figure 2.** Mapping D-loops formed on a supercoiled donor by DMA. (A, B, C) Footprint maps depicting reads with a D-loop footprint. Reads were derived from an in vitro D-loop reaction performed with a supercoiled donor and ds98-*931*, ds98-*607*, or ds98-*197*-78ss substrates, respectively. Only reads from the top strand of dsDNA donor that contains a footprint are shown here. Here and in all subsequent figures with a footprint map, each horizontal line represents one read molecule (or amplicon). Vertical yellow lines indicate the position of each cytosine across the read sequence. The status of each cytosine along the sequence is color-coded with green representing C-T conversions. The status of cytosine is changed to red if the C-T conversions cross the peak threshold and are defined as D-loop footprints. Unless otherwise mentioned, the peak threshold is t40w50 (requiring at least 40% cytosines converted to thymine in a stretch of 50 consecutive cytosines). The reads are clustered based on the position of footprints in a 5' to 3' direction. The faintly colored boxes indicate the clusters. The blue dotted box represents region homologous to the invading substrate. The text below the map summarizes the number of reads containing a footprint (depicted in the map) and the total top-strand reads analyzed for that sample. Some of the footprints seem to extend slightly beyond the region of homology. This may be due inclusion of converted cytosines from DNA breathing close to the D-loop. Scale bar is 100 nt. (D) Table summarizing the total number of reads containing a footprint as 'peak' and the total number of reads analyzed as 'total' for each strand. '% Peak' indicates the percentage of reads containing a footprint calculated by dividing the number of reads containing a footprint by the total number of reads for that strand. The data represents a cumulation from >3 independent replicates. (E) Quantitation of D-loops from the gel in *Figure 2—figure supplement 1* is compared to the percentage of reads with D-loop footprint, as observed by DMA. Mean ± SD (n = 3).

The online version of this article includes the following figure supplement(s) for figure 2:

**Figure supplement 1.** Gel-based assay for in vitro formed D-loops.

levels detected by DMA, the relative D-loop levels were comparable across the two orthogonal D-loop detection assays (*Figure 2E*).

## Position of D-loops formed by a supercoiled donor is restricted within the homology window, while their distribution reveals an enrichment at the 3'-end of homology

Next, we looked at the position of individual D-loops relative to the dsDNA sequence. Most of the D-loop footprints were confined within the 931 nt region of homology (depicted by a blue box in the footprint map) for the ds98-*931* substrate (*Figure 2A*). As we reduced the length of homology shared between the ssDNA substrate and the dsDNA donor, the position of the D-loop footprints changed to reflect this differential homology length (*Figure 2A–C*). Even with a ds98-*197*-78ss substrate having only 197 nt homology to a 3,000 bp donor DNA, all the D-loop footprints were confined within the 197 nt homology window. Thus, the D-loop footprints were specific to homology, irrespective of the length of homology.

Next, we examined the distribution of the D-loop position within the homology window for each substrate type. As evident from the visual appearance of D-loop footprints clustered based on their position on the footprint map, the D-loops seemed enriched at the 3′-end of homology (*Figure 2A–C*). We quantified the distribution of D-loop position as the fraction of D-loops found within 100 nt bins non-exclusively. The distribution of D-loops across the homology length for ds98-*931* and ds98-*607* substrate is shown in *Figure 3A and B*, respectively. Similar to the results from a restriction-digest based assay (*Wright and Heyer, 2014*), there was an enrichment of D-loops near the 3′-end of ssDNA, while the 5′-end of homology had the lowest D-loop presence. This observation may support the model that Rad51 filament grows preferentially in a 5′-to-3′ direction (*Qiu et al., 2013*; *Špírek et al., 2018*), making it more likely for the D-loops to assemble at the 3′-end of homology than at the 5′-end. Only D-loops in which the 3′-OH end is incorporated into the hDNA are competent for D-loop extension (*Li and Heyer, 2009*). Alternatively, the enrichment of D-loops at the 3′-

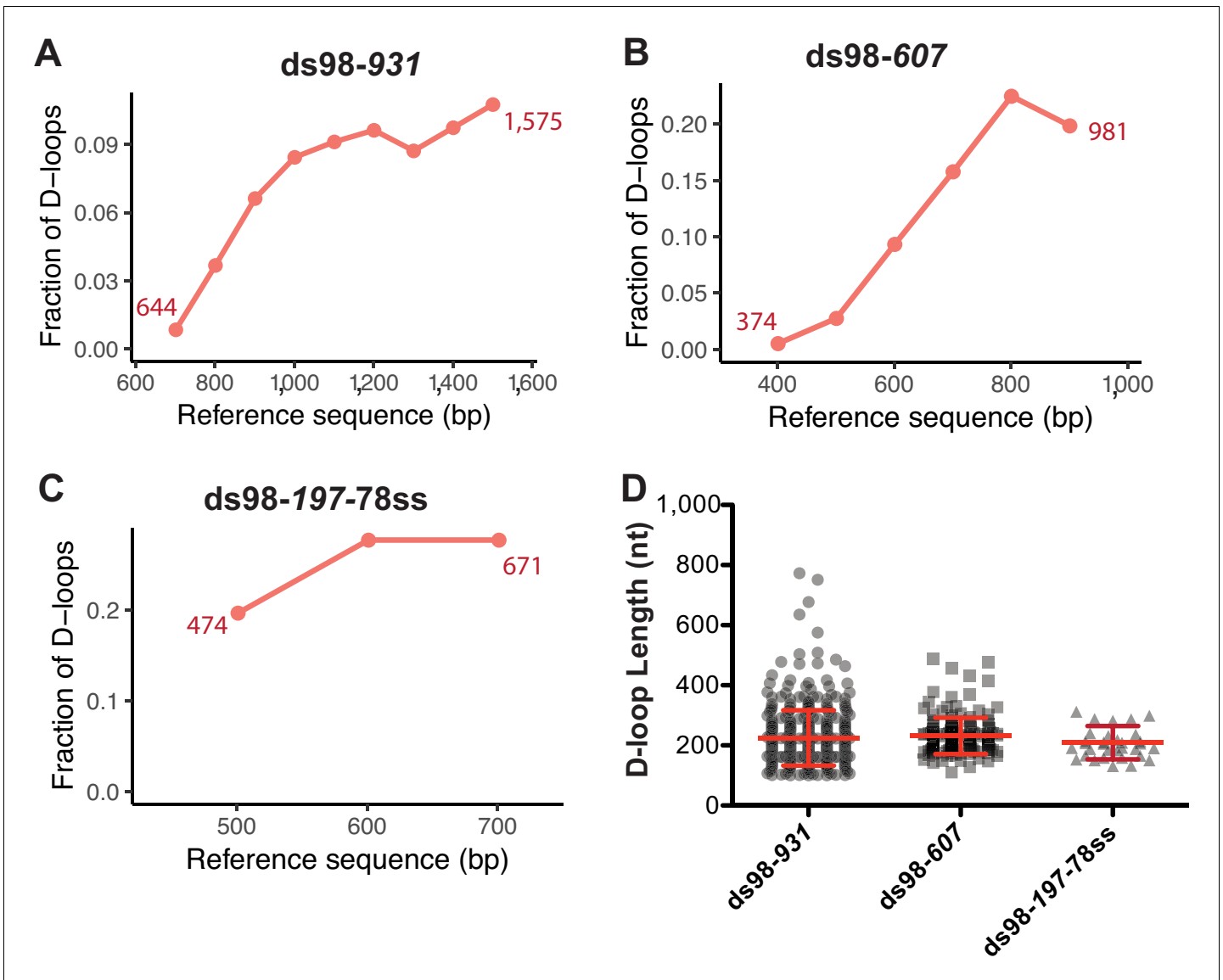

**Figure 3.** Distribution of D-loop lengths and position, when formed on a supercoiled donor. (A, B, C) Distribution of D-loop footprints across the region of homology for the ds98-*931*, ds98-*607*, or ds98-*197*-78ss substrates, respectively. Here, and in all subsequent graphs, the distribution is measured by binning each footprint in 100 nt bins across the homology non-exclusively. The numbers by the plotted line represent the beginning and end of the exact region of homology for each substrate type on the reference sequence. (D) Dot plot showing the distribution of D-loop lengths seen with each substrate type in the DMA assay. In red is Mean ± SD (n = 3).

end might be due to the higher physical flexibility of ssDNA at the 3′-end compared to the 5′-end that is near a duplex DNA. Lastly, for the ds98-*197*-78ss substrate, due to the small window size of only 197 nt, the distribution of D-loop position is less informative, but is shown in *Figure 3C*.

Note that a slight drop in the frequency of D-loops was seen in the last ~100 nt of homology near the 3′-end for the ds98-*931* and ds98-*607* substrates (*Figure 3A,B*). This drop in the frequency could be due to end bias in calling a D-loop footprint and/or in binning the D-loop coverage. Since no D-loop is expected after the last nucleotide of homology, there would be fewer, if any, cytosine conversions after the 3′-end of homology. Hence, the window of peak threshold used may be slightly smaller, not including the last cytosine as often as it should. Additionally, lack of D-loops at the other side of the homology end may create a slight drop in the coverage frequency with 100 nt bins. However, the overall trend is not expected to change drastically. In summary, for long homology windows, a distribution of D-loop coverage provides insight towards biases in the D-loop position.

## Length of D-loops with a supercoiled donor is restricted by the supercoiling density of the donor

To determine the length of individual D-loops in nucleotides, we measured the length of the footprints. The precise margins of a D-loop footprint are influenced by a combination of the peak threshold used to define a footprint and the bisulfite conversion efficiency (see later discussion on peak thresholds).

As stated earlier, with a supercoiling density of ~20 ± 5 in the donor DNA, D-loops are expected to have a maximum length of ~210 ± 50 nt, irrespective of the homology length. We observed that D-loop footprints have an average length of 259 ± 93 nt and 231 ± 60 nt distributed across the region of homology with the ds98-*931* and ds98-*607* substrates, respectively (*Figure 3D*). With the 197 nt homology substrate (ds98-*197*-78ss), the average D-loop length was 206 ± 48 nt (*Figure 3D*). Thus, when using a super-coiled donor, D-loop length is tightly regulated by the supercoiling density of dsDNA irrespective of the size of homology between the substrate and the donor.

Note that these average lengths are slightly higher than the maximum expected length of ~210 ± 50 nt for ds98-*931* and ds98-*607* substrates. This slightly higher average length is due to a small fraction (<5–10%) of footprints that were much longer than expected (>350 nt) with a supercoiled donor (*Figure 3D*, *Figure 2—figure supplement 1D*). These longer D-loops were nevertheless still within the maximum possible length based on the region of homology that is shared between the ssDNA substrate and the supercoiled duplex donor. Such long D-loop footprints may either be due to a small fraction of nicked (relaxed) plasmids in the reaction or due to branch migration of D-loops during the bisulfite treatment. Since the D-loops formed on supercoiled donors are very stable at room temperature (*Wright and Heyer, 2014*), branch migration appears to be the less likey explanation. Additionally, separation of the D-loop reactions on a gel revealed that 5–10% of the supercoiled plasmid was nicked/relaxed (*Figure 2—figure supplement 1A–B*). Nicked plasmids would have unrestricted topology, allowing the formation of D-loops that span the entire region of homology. Thus, we conclude that these longer D-loops are the result of nicked dsDNA molecules. Since most of the footprints had the D-loop length expected from the topological limit, it seems that branch migration, if at all, was minimal on supercoiled donors. This suggests that no significant changes in D-loop length or position occur during the bisulfite treatment. For the ds98-*197*-78ss substrate the average D-loop length of 206 nt was slightly larger than the 197 nt length of homology. The 197 nt homology comprises only 43 cytosines, lower than the required minimum of 50 cytosines with a t40w50 threshold. With a conversion rate greater than 40%, surrounding cytosines maybe incorporated in a peak, resulting in slight overestimation of D-loop length when the homology length is close to the threshold limit. In fact, with a t40w40 threshold, the average D-loop length was 191 ± 41 nt.

In summary, the lack of footprints outside the region of homology, their absence on the bottom-strand, their dependence on the formation of a D-loop, and their uniform length restricted by the supercoiling density of the donor provide high confidence that the footprints represent D-loops.

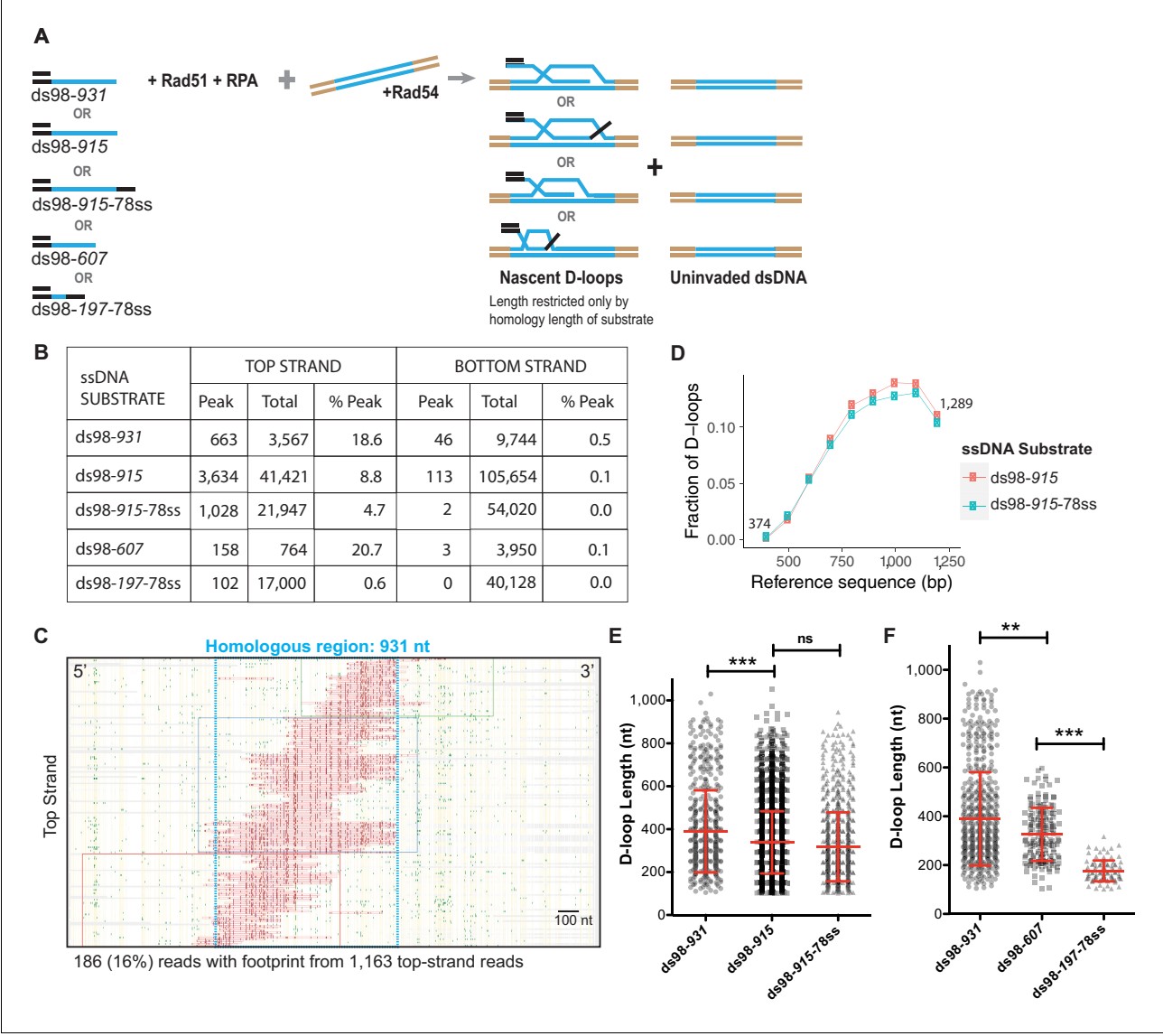

**Figure 4.** Characteristics of the D-loops formed on a linear donor. (**A**) Schematic of the in vitro D-loop reaction involving linear dsDNA donor and various substrates. The blue lines indicate the homology between the substrate and the donor. These substrates can form different D-loops depending on their homology size, homology position, and flanking heterology. Linear donors lack topological restrictions. (**B**) Table summarizing the total number of reads containing a footprint as 'peak' and the total number of reads analyzed as 'total' for each strand. '% Peak' indicates the percentage of reads containing a footprint. The data represents a cumulation from >3 independent replicates. Refer to *Figure 4—figure supplement 1*. (**C**) Footprint map of D-loop sample formed with ds98-*931* and a linear donor. Refer to *Figure 4—figure supplement 2* for other footprint maps. (**D**) Comparing the distribution of D-loop position between an invading substrate with (ds98-*915*-78ss) and without (ds98-*915*) a non-homologous 3′-end (n = 3). (**E**) Distribution of D-loop lengths observed for the three substrates with ~900 nt homology to the donor. Mean ± SEM in red (n = 3). (**E**) Dot plot showing a distribution of D-loop lengths seen with substrates having ~900 nt homology to the donor in the DMA assay. In red is Mean ± SD (n = 3). *** indicates p-value<0.0005, ns indicates non-significant using a two-tailed, Student's t-test. (**F**) Dot plot showing a distribution of D-loop lengths seen with substrates having different lengths of homology to the donor in the DMA assay. In red is Mean ± SD (n = 3). ** indicates p-value<0.005, ***<0.0005, using a two-tailed, Student's t-test.

The online version of this article includes the following figure supplement(s) for figure 4:

**Figure supplement 1.** Quantification of D-loops formed on a linear donor from the gel-based assay and DMA.

**Figure supplement 2.** Characterization of D-loops formed on a linear donor by DMA.

**Figure supplement 3.** Characterization of D-loops formed using human recombinant proteins by DMA.

## Characterization of D-loops formed on a linear donor reveal enrichment of D-loops at the 3′-end and depict a wide range of D-loop lengths across the region of homology

After establishing the assay using a supercoiled donor, a linearized plasmid, free of topological constraints, was used as a donor to map the D-loops in a topologically uninhibited manner. Again, a range of ssDNA substrates was used with varying lengths of homology to the donor (ds98-*915*, ds98-*607*, ds98-*197*-78ss), different sequences (ds98-*931 vs.* ds98-*915*) or presence of a non-homologous 3′-flap (ds98-*915*-78ss) (*Figure 4A*). Substrates with different homology lengths with respect to the donor were used to determine whether this would result in corresponding changes in D-loop length distribution. While substrates with similar homology length but different homologous sequences to the donor were used to test for any sequence bias contributing to the D-loop distribution. The ds98-*931* and ds98-*915* substrates differ in sequence by >33%, and the sequences are provided in the Key Resources Table. Lastly, the substrate with a non-homologous 3′-flap was used to test if 3′-heterology affects the enrichment of D-loops at the 3′-end of homology.

As expected, the D-loop footprints for each of these substrate types were specifically seen only on the top strand, as evident in the cumulative read data (*Figure 4B*). A total of ~800–3,700 (10–18%) D-loops were observed among ~4,000–40,000 top-strand reads (*Figure 4B*). Again, while the bottom-strand comprised ~13,000–100,000 reads, it had <0.1% of reads with a footprint. (Here, a 10-fold difference in the total number of reads analyzed was due to upgrade in Pacific Biosciences sequencing system, from Sequel -I to Sequel – II. See Material and methods). Thus, even with a linear, topologically unrestricted donor, the D-loops were mapped with high strand-specificity (>100 fold). Relative D-loop levels quantified from DMA correlated with the D-loop levels from the gel-based assay (*Figure 4—figure supplement 1A,B*). Although, as expected from previous observations with negatively supercoiled donors, the D-loop levels estimated from DMA were lower than the ones from the gel.

Moreover, the D-loop footprints were essentially all found within the window of homology for each substrate type, as evident from individual footprint maps (*Figure 4C*, *Figure 4—figure supplement 2A–C*), attesting to the homology-specificity of the footprints. In some cases, less than 0.4% of footprints were found outside the region of homology (*Figure 4—figure supplement 2B*). These footprints were found on both the top strand and bottom-strand reads, attesting their origin to be due to DNA breathing in a C-rich region. Next, the distribution of the D-loop position with a linear donor revealed an enrichment at the 3′-end of homology (*Figure 4D*, *Figure 4—figure supplement 2D,E*), similar to the D-loops formed using a supercoiled donor. Moreover, the presence of a non-homologous 3′-flap in ds98-*915*-78ss sample did not significantly alter the distribution of the D-loop position (*Figure 4D*) compared to the ds98-*915* sample. Despite the flap, the D-loops were enriched at the 3′-end of homology in a similar fashion. Since the non-homologous flap is only 78 nt long, it may not alter the distribution of Rad51 filament significantly, resulting in similar D-loop allocations. It may be interesting to test if a non-homologous flap longer than the average D-loop length of 450 nt has any effect on D-loop distribution.

The distribution of D-loops was also unaltered by changes in the homology sequence between the ds98-*931* and ds98-*915* substrates (*Figure 4D*, *Figure 4—figure supplement 2D*). This suggests that differences in cytosine distribution across the homology do not significantly alter D-loop distribution, as long as the % CG is not very low. In these cases, the % CG were >40%. Thus, these observations further support the 5′-to-3′ directionality of the Rad51 filament growth (*Qiu et al., 2013*; *Špírek et al., 2018*).

Interestingly, there was no preferred D-loop length. Instead, a broad range of D-loop lengths was observed, independent of the substrate type (*Figure 4E,F*). The lengths of the D-loops ranged from ~100 nt to a maximum spanning the entire length of homology. D-loops as long as 900 nt were seen with the long substrates (ds98-*915,* ds98-*931*, and ds98-*915*-78ss) at a frequency of ~5% of the total D-loops (*Figure 4E*). The minimum D-loop size of 100 nt is limited by the peak threshold (of t40w50) used to define a D-loop footprint. An average D-loop length of ~390 nt was noted with the 900 nt homology series substrates. There was a slight, yet significant difference in the D-loop lengths between the ds98-*915* and ds98-*931*, with a difference in mean of 60 nt. Two factors may contribute to this difference in D-loop lengths. One, the 16 nt longer homology in ds98-*931* may contribute to relatively longer average D-loop length. Second, the two substrates differ in their homologous

sequences. The ds98-*915* substrate includes 192 cytosines, while the ds98-*931* has 187 cytosines within the region of homology. Thus, the relatively less frequent cytosine distribution in ds98-*931*, despite the longer homology length, may consequently result in the observed increase in the mean D-loop length. However, there was no statistical difference in D-loop lengths between the ds98-*915* and ds98-*915*-78ss substrates, suggesting that the 78 nt 3′-flap does not alter D-loop lengths. Overall, >40% of footprints were longer than the average length of 400 nt for each of these three substrates. Thus, the differences in the homology sequence or the presence of a non-homologous 3′-flap did not significantly alter the distribution of D-loop lengths.

As expected, the average D-loop length decreased from ~390 nt to ~320 nt, when the homology length was reduced to 607 nt using the ds98-*607* substrate (*Figure 4F*). The maximum D-loop length for the ds98-*607* substrate was 600 nt, restricted only by the homology length. About 6% of the D-loops were longer than 500 nt, traversing across the region of homology. Similarly, the D-loops formed with the ds98-*197*-78ss substrate had an average length of 176 nt, and ~73% spanned the entire region of homology. The minimum D-loops length observed was ~100 nt limited by the peak threshold of t40w50. In summary, our results demonstrate that homology length defines the maximum observable D-loop length, and in turn, influences the average D-loop length. Note that since the D-loops formed on a linear donor are relatively less stable than those on a supercoiled donor (*Wright and Heyer, 2014*), it is possible that the footprints may reflect all changes in D-loop length and position that occurred over the course of the bisulfite treatment.

Formation of multi-invasions, where a single substrate forms D-loops simultaneously in multiple donor molecules, was shown both in vitro and in vivo (*Wright and Heyer, 2014*; *Piazza et al., 2017*). However, multi-invasions cannot be detected by this assay, since each donor molecule is separately analyzed. Instead, in some cases (~2–5%), multiple D-loop footprints are seen within a single read (*Figure 4—figure supplement 2A–C*), representing a single donor. This may be due to multiple invasions of one or more substrates into a single donor at two different sites (*Figure 4—figure supplement 2F*). Multiple invasions from a single substrate at two sites within a donor may arise due to the formation of two Rad51 filaments separated by a gap. However, Rad54 may be able to translocate through such a gap, making these kinds of multiple invasions a rarity. Alternatively, two substrates invading a single donor may also give rise to dual footprints. Conversely, two D-loop footprints on a single read may also arise due to inefficient cytosine conversions across a long D-loop, possibly due to the continued presence of RPA on the displaced strand. Unless the two footprints are separated by >200 nt (~50 cytosines), it is currently hard to distinguish between the two possibilities.

## RAD51 and RAD54 also form D-loops of varying lengths similar to Rad51 and Rad54 and depict an enrichment of D-loops at the 3′-end

We also similarly tested the D-loop mapping assay on D-loops formed using the human recombination proteins RAD51, RAD54, and RPA. The rationale for using human proteins was to test whether the D-loop length and distribution was an inherent property of the yeast Rad51 recombinase. In vitro D-loop reactions with human proteins were performed using the ds98-*931* substrate and a supercoiled donor (*Figure 4—figure supplement 3A*) or a linearized donor (*Figure 4—figure supplement 3C*). With both donor types, the D-loop footprints were again specifically seen within the homologous region (*Figure 4—figure supplement 3B,D*) and on the top strand (*Supplementary file 1*). The distribution of D-loops was similar to the D-loops produced by the yeast proteins. An enrichment of D-loops was seen at the 3′-end of homology, irrespective of the donor type (*Figure 4—figure supplement 3E*). Average D-loop length seen with a supercoiled donor was 252 ± 78 nt as expected, while with a linear donor, the average length was 320 ± 140 nt (*Figure 4—figure supplement 3F*). The average D-loop length from a linear donor was slightly lower than the 410 nt seen with yeast proteins. Conformingly, only 21% of the D-loops were longer than 400 nt with human recombinant proteins compared to >40% seen with yeast recombinants. Only one D-loop spanned the entire length of homology. Thus, compared to the yeast proteins, human proteins seemed to be less efficient under the conditions used in forming longer D-loops. As evident from observation of the D-loops on an agarose gel (*Figure 4—figure supplement 3G,H*), the efficiency of D-loop formation was also much lower, with only 3% using a linear donor. This could be due to the poor stability of the RAD51 filament. It may also suggest a role for RAD51 accessory factors in stabilizing the RAD51 filament and altering its properties to promote more efficient D-loop

formation (*San Filippo et al., 2008*). Concomitantly, the percentage of D-loops observed on a gel correlated well with the frequency of D-loop footprint (*Figure 4—figure supplement 3H*). Thus, the D-loop Mapping Assay can be broadly used to test the effect of recombinant proteins across various species on D-loop properties.

## Optimization of the D-loop Mapping Assay

### Temperature conditions for bisulfite treatment

The bisulfite conversion is most efficient at 70℃ for 30 min (*Yi et al., 2017*). Such high temperatures would be deleterious for our purpose as it would cause D-loop dissolution (*Wright and Heyer, 2014*) or increased DNA breathing. We, therefore, varied temperature and time conditions to minimize background conversion from DNA breathing and to prevent branch migration and dissolution of D-loops, while retaining good bisulfite conversion efficiency. We tested the following temperature and time combinations: 37℃ for 30 min, 37℃ for 4 hr, 30℃ for 1 hr, RT (25℃) for 3 hr, RT for 4 hr, while keeping everything else constant. A D-loop reaction performed in the absence of ssDNA substrate was used as a negative control to test for non-specific conversions. The bottom-strand of dsDNA and the 1,000 bp of non-homologous sequence on top strand also acted as internal negative controls.

A 4 hr incubation at 37℃ showed cytosine conversion patches even outside the homologous region and on the bottom-strand (*Figure 5—figure supplement 1A*), indicating enhanced DNA breathing at 37℃ (note that the number of reads analyzed was lower as the lower throughput RS-II sequencing platform was used). Incubation at 37℃ is also known to enhance D-loop migration and dissolution (*Wright and Heyer, 2014*). Bisulfite treatment at 37℃ for 30 min and at RT for 3 hr showed better recovery of D-loops compared to other treatment conditions (*Figure 5—figure supplement 1A–C*), depicting a higher fraction of reads with a D-loop footprint. To minimize potential D-loop migration, we opted for a 3 hr incubation at RT as the optimal condition for the bisulfite treatment.

### Defining a peak threshold – t40w50 is an optimum peak threshold

A footprint is called based on a predefined peak threshold that can be altered in the analysis pipeline. A peak threshold is a minimum conversion frequency (t) required within a minimum window size (w) of consecutive cytosines to call a footprint. The 't' parameter in a peak threshold should be lower than the average conversion efficiency of bisulfite treatment to minimize false-negative footprints. Conversely, the 'w' parameter is set to eliminate false positives from spontaneous DNA breathing.

An ssDNA (pBSKS- circular DNA) was spiked into an in vitro D-loop reaction sample to measure the efficiency of cytosine conversion. 58% of the 420 cytosines were converted on average in each individual read, and 100% of the 21 reads had cytosine conversion (*Figure 5—figure supplement 2A,B*). The cumulative cytosine conversion efficiency was thus, 58%. This finding implies that choosing a 't' value lower than 58% should prevent false elimination of true footprints. Note that the conversion efficiency will always be less than 100% since non-denaturing conditions are applied.

To optimize the 'w' parameter, we examined spontaneous DNA breathing in the dsDNA (uninvaded) sample along with an in vitro D-loop sample. Footprints arising from spontaneous DNA breathing would be present on both the top and bottom-strands of the dsDNA, as well as in the regions lacking homology to the ssDNA substrate. We tested two window sizes of 20 and 50 consecutive cytosines with a range of 't' parameters, 25, 30, 40, and 60%, hereby denoted as t25w20, for example (*Figure 5A*, *Figure 5—figure supplement 2B,C*). For a D-loop sample, with peak thresholds of t40w50 and t60w20, almost no footprints were observed on the bottom-strand or in the flanking non-homologous regions (*Figure 5A*, *Figure 5—figure supplement 2C*). With t40w20, 1.4% of bottom-strand reads had such non-specific footprints. Lowering the conversion frequency (t) to 25% significantly increased footprints on the bottom-strand, along with a concomitant increase in the percent of footprints detected on the top strand. 11.6% and 8.8% of bottom-strand reads had a footprint with t25w20 and t25w50, respectively, compared to ~40% top-strand footprints (*Figure 5A*, *Figure 5—figure supplement 2C*). We conclude that at least a t40 conversion threshold is required to eliminate the majority of footprints arising from spontaneous DNA breathing. To further eliminate false positives among the D-loop footprints, we decided on the next least stringent condition of t40w50 with no detectable non-specific footprints. While t60w20 also eliminated

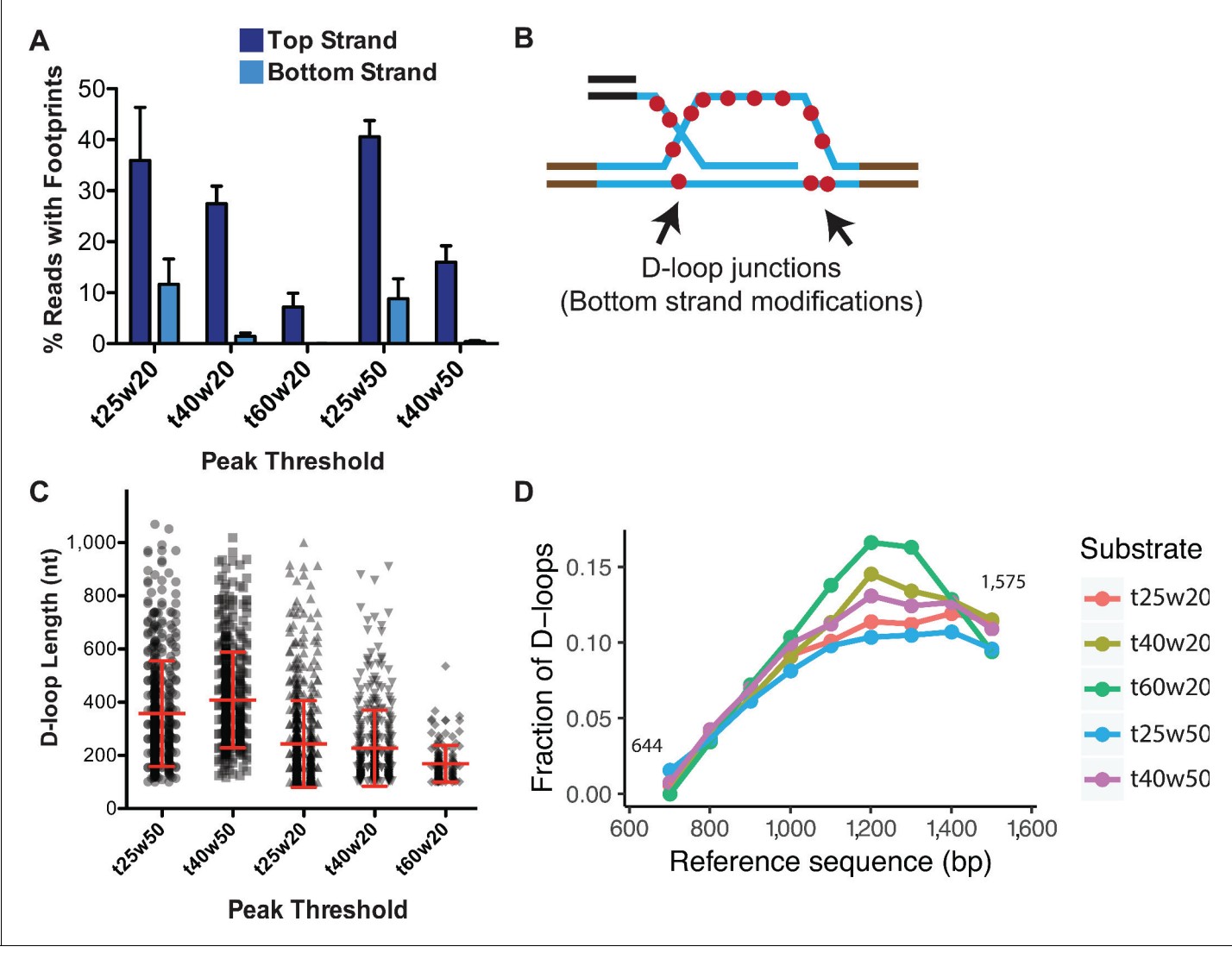

**Figure 5.** Effect of varying peak thresholds on D-loop levels, length, and position. (**A**) Percentage of reads with footprint when called with different peak thresholds for D-loops formed with ds98-*931* and a linear donor. For each peak threshold, 't' represents the minimum conversion frequency of cytosines, 'w' represents the window size requiring the minimum number of consecutive cytosines. Refer to *Figure 5—figure supplement 2*. (**B**) Schematic representation of potentially short, bottom-strand footprints derived from the D-loop junctions. (**C**) Changes in the distribution of D-loop lengths as defined with different peak thresholds. D-loops formed with ds98-*931*, and a linear donor are depicted. Mean ± SD (n = 3) in red. (**D**) Distribution of the position of D-loop footprints when defined with different peak thresholds.

The online version of this article includes the following figure supplement(s) for figure 5:

**Figure supplement 1.** Optimizing bisulfite treatment conditions.
**Figure supplement 2.** Optimizing the peak threshold for DMA.

background footprints, the frequency 't' requirement was higher than the average conversion frequency seen. Hence, most of the genuine top-strand D-loop footprints were lost, reducing the percentage of D-loops to 8% (*Figure 5A*). Thus, with respect to specificity, the t40w50 peak threshold is the most optimal for detecting D-loop footprints.

Interestingly, while the t40w50 threshold eliminates the majority of bottom-strand footprints, at the t25w20 threshold, a large number of bottom-strand reads showed two footprints within a read (*Figure 5—figure supplement 2B*). These two short footprints within a single read may have arisen at the two junctions of a D-loop, where the bottom-strand may be single-stranded (*Figure 5B*). RPA may bind at the 3′ junction of a D-loop (*Sneeden et al., 2013*), indicating that at least 25 nt (the binding site size of RPA) single-stranded region may be present on the bottom-strand. However, it

is currently not possible to determine if they truly represent D-loop junctions, or simply arise from breathing of the DNA. The footprints outside the region of homology on the bottom-strand surely arise from the breathing of the DNA.

## Effect of different peak thresholds on D-loop length and distribution

While optimization of peak threshold helped minimize false positives and false negatives, we wondered if it had any effect on the overall D-loop features. To address the effect of peak thresholds on D-loops called, we examined D-loop lengths and distribution under the various peak threshold conditions with the ds98-*931* substrate and a linear donor.

In terms of the D-loop lengths, as one may expect, more short-length footprints were called with the low stringency conditions. With t25w50, 22% of the footprints were shorter than 250 nt, compared to 5% with a t40w50 threshold. Consequently, the average D-loop length with a t25w50 threshold was 357 nt compared to 410 nt with a t40w50 threshold (*Figure 5C*). Additionally, lowering the window size from w50 to w20 further lowered the average D-loop length. The average length was 220 nt for t40w20, with 59% of the D-loops being shorter than 200 nt. Thus, with lower window size, more of the short D-loop footprints were captured along with some footprints potentially arising from spontaneous DNA breathing, leading to an overall reduction in average D-loop length.

Note that genuine-D-loops shorter than 250 bp would be most affected by the window size. This is because a window size of 20 would result in a minimum D-loop size of 80–100 bp, considering a CG distribution of >40%. While using a window of 50, the minimum D-loop size would be 120–200 bp. Hence, for w = 50, D-loops that were originally 180 bp would be slightly overestimated in size as 200–250 bp, assuming that the cytosine conversion frequency permits it. On the other hand, longer D-loops will be unaffected by the window size. The length of genuine D-loops will also be unaffected by a change in conversion frequency below the average bisulfite conversion frequency.

Lastly, changes in peak threshold did not affect the overall distribution of D-loops within the homology window (*Figure 5D*). Thus, despite the detection of short non-specific footprints at lower thresholds, there was no bias in the position of these non-specific footprints, meaning that there was no overall change in their distribution.

## Discussion

We have developed a novel D-loop mapping assay to determine the length, position, and distribution of in vitro formed D-loops for the first time. The method provides information on individual D-loop molecules with almost base-pair precision. The method is high-throughput, sensitive, and readily applicable to various kinds of D-loops. It can be broadly used to test the effect of various homologous recombination factors on D-loop characteristics. It is also possible to extend the assay to map other DNA intermediates/structures that include some regions of ssDNA. Using the D-loop Mapping Assay we could show that Rdh54/Tid1 antagonizes Rad54 in D-loop formation and restricts D-loops length (*Shah et al., 2020*).

Using the novel DMA, we have shown that without a topological barrier, D-loop length can vary widely, in some cases stretching as long as the homology permits. These differences in D-loop length may either reflect differential Rad51 filament lengths or variations in the processivity of Rad54 in forming hDNA. Moreover, the D-loops are enriched at the 3′-end of homology, similar to previous observations (*Wright and Heyer, 2014*). This enrichment confirms the directionality of the Rad51 filament in a 5′-to-3′ direction (*Qiu et al., 2013*). The 3′ enrichment may also suggest that the branchpoint with the 3′-OH end of ssDNA better activates the motor activity of the Rad54 translocase. Furthermore, the presence or absence of a topological barrier also does not alter the distribution of the D-loop position across the region of homology. The D-loops are enriched at the 3′-end, irrespective of topological barriers or D-loop size. Lastly, the presence of a 78 nt non-homologous 3′-flap has no significant effect on the distribution of D-loop length or position. It may be interesting to test if a longer heterology at the 3′ end would alter the distribution of D-loop position and/or length. Thus, DMA can provide valuable insights on D-loop characteristics, that may help reveal the roles of D-loop modulators and their regulation of D-loop dynamics.

## Advantages and limitations of the D-loop mapping assay

The D-loop Mapping Assay provides the following advantages:

1. DMA provides a tool to determine, for the first time, the lengths of individual D-loops and their distribution in a population of D-loops formed in vitro with near base-pair resolution.
2. DMA allows mapping the exact position of each D-loop in reference to the donor dsDNA sequence. Distribution of the D-loop position can reveal any biases in the position of a D-loop within a region of homology. For instance, it revealed that D-loops are more frequently formed at the 3′-end of the homology.
3. Relative D-loop levels detected by DMA are comparable across the different D-loop samples.
4. The in vitro DMA is backed by several controls that allow testing the specificity of the assay within each sample and also within each read molecule.
5. There is no bias in the length of the D-loop mapped by the DMA above the 120–200 nt threshold limit. The bias is avoided due to the lack of a D-loop enrichment step or a D-loop crosslinking step. Hence, short and long D-loops are mapped with similar frequencies as evident from the vast and equal distribution of D-loop lengths on a linear donor.
6. DMA is a highly sensitive method to detect and map D-loops. DMA can map D-loops formed with efficiency as low as ~1–2% using Sequel-I or as low as ~0.05% with the Sequel-II system (higher coverage).
7. DMA allows multiplexing of samples with up to 50–200 barcoded samples analyzed simultaneously. This provides speedy and cost-efficient high-throughput analysis.

The D-loop mapping assay is limited in the following ways:

1. The minimum detectable D-loop length is limited by the peak threshold used for the analysis of DMA. The peak threshold is used to ensure confidence in that a conversion patch is derived from a genuine D-loop and not spontaneous breathing of the duplex DNA. D-loops shorter than 120–200 nt are undetectable with a threshold window of 50.
2. There is lower confidence in the precise length of those D-loops whose length approaches or is lower than the threshold window. This is because the length may be slightly overestimated due to the requirement of a minimum threshold window. However, D-loop with a length longer than the threshold window is not affected.
3. As with any D-loop assay, some unstable D-loops may be lost during the bisulfite treatment. Some D-loop molecules may also be lost due to the nicking of ssDNA during bisulfite treatment. Despite this, due to the lack of experimental bias, D-loops can be compared across different samples.
4. Some long D-loops may be misidentified as short D-loops due to insufficient bisulfite conversion within the treatment time period. The insufficient bisulfite conversion might be due suboptimal room temperature treatment condition or due to incomplete RPA removal by proteinase K. However, optimizing the conversion threshold by using a spiked ssDNA control helps minimize the possibility of underestimated D-loop lengths or false negatives.
5. Some short D-loops may be misrepresented as long footprints due to any potential D-loop migration during the course of bisulfite treatment. However, D-loops are presumed to be relatively stable at room temperature, as no D-loop dissolution is observed over time (*Wright and Heyer, 2014*). Moreover, most of the footprints representing supercoiled D-loops had the expected size, minimizing the possibility of branch migration, especially on supercoiled donors.

Lastly, we suggest some potential technical alterations to the DMA that may improve the conversion efficiency or the D-loop detection levels.

1. The proteinase K digestion and removal of RPA from the displaced strand may be less efficient without the presence of a detergent. To minimize the possibility of RPA bound on the displaced strand during bisulfite conversion, we suggest testing the use of CTAB detergent along with proteinase K for any potential improvement in the D-loop footprint signal. CTAB, unlike SDS, is known to prevent D-loop migration (*Allers and Lichten, 2000*).
2. Currently 35 cycles are performed during each round of PCR. The high cycle number may be promoting the 2-fold bias currently observed among the top and bottom-strand reads. Reducing the cycle number during PCR-1 may help reduce the strand bias. However, these are minor changes and may not significantly change the overall footprint map or relative comparison across different samples.

# Materials and methods

**Key resources table**

| Reagent type (species) or resource | Designation | Source or reference | Identifiers | Additional information |
|---|---|---|---|---|
| Recombinant DNA reagent (plasmid) | pBSKS (-) strand | *Wright and Heyer, 2014* | | Used to test bisulfite conversion efficiency |
| Recombinant DNA reagent (plasmid) | pBSphix1200 | *Wright and Heyer, 2014* *Wright and Heyer, 2014* | Amp | Used as dsDNA donor in D-loop assay in supercoiled or linear form |
| Recombinant protein (*S. cerevisiae*) | Rad54 | *Wright and Heyer, 2014* | | |
| Recombinant protein (*S. cerevisiae*) | Rad51 | *Van Komen et al., 2006* | | |
| Recombinant protein (*S. cerevisiae*) | RPA | *Binz et al., 2006* | | |
| Oligonucleotide | UNI+Donor-PB-F | This paper | | GCAGTCGAACATGTAGCTGACTCAGGTCACTCACACTTCCTGGTTGATGG |
| Oligonucleotide | UNI+ PhiX-PB-R | This paper | | TGGATCACTTGTGCAAGCATCACATCGTAGATCTACACGACGGGGAGTCA |
| Oligonucleotide | pBS-915nt-subs-R | This paper | | GGTATCGATAAGCTTCCATG gcatttgtttcagggttatttg |
| Oligonucleotide | pBS-51-931nt-subs-F | This paper | | CGTATCTAGACTGCA gaacggaaaacatccttcatag |
| Oligonucleotide | pBS-1013nt subs-R | This paper | | CGTATCTAGACTGCA gaagtcatgattgaatcg |
| Oligonucleotide | pBS-1013nt subs-F | This paper | | GGTATCGATAAGCTTCCATG gttaatgccactcctctcccga |
| Oligonucleotide | 3'-non-tailed-NcoI-915 | This paper | | GATAAGCTTCCATGGCAT |
| Oligonucleotide | UNI+pBSKS-F | This paper | | GCAGTCGAACATGTAGCTGACTCAGGTTTTTGATTTATAAGGGATTTTG |
| Oligonucleotide | UNI+pBSKS-R | This paper | | TGGATCACTTGTGCAAGCATCACATCGTAGTTTATTTTTCTAAATACATTCAAATAT |
| Oligonucleotide | *100*-mer | *Wright and Heyer, 2014* | | ctggtcataatcatggtggcgaataagtacg cgttcttgcaaatcaccagaaggcggttcctg aatgaatgggaagccttcaagaaggtgataagcagga |
| Oligonucleotide | ds98-*197*-78ss | *Wright and Heyer, 2014* | | Homologous sequence (197 nt): ctggtcataatcatggtggcgaataagtacg cgttcttgcaaatcaccagaaggcggttcctg aatgaatgggaagccttcaagaaggtgataa gcaggagaaacatacgaaggcgcataacga taccactgaccctcagcaatcttaaacttctta gacgaatcaccagaacggaaaacatcctt catagaaattt |

*Continued on next page*

*Continued*

| Reagent type (species) or resource | Designation | Source or reference | Identifiers | Additional information |
|---|---|---|---|---|
| Oligonucleotide | ds98-607 | *Wright and Heyer, 2014* | | Homologous sequence (607 nt): gaagtcatgattgaatcgcgagtggtcgg cagattgcgataaacggtcacattaaattt aacctgactattccactgcaacaactgaa cggactggaaacactggtcataatcatgg tggcgaataagtacgcgttcttgcaaatca ccagaaggcggttcctgaatgaatgggaagc cttcaagaaggtgataagcaggagaaacatac gaaggcgcataacgataccactgaccctcagcaa tcttaaacttcttagacgaatcaccagaacggaaa acatccttcatagaaatttcacgcggcggcaagttgcc atacaaaacagggtcgccagcaatatcggtataagt caaagcacctttagcgttaaggtactgaatctctttagt cgcagtaggcggaaaacgaacaagcgcaagagtaa acatagtgccatgctcaggaacaaagaaacgcggca cagaatgtttataggtctgttgaacacgaccagaaaa ctggcctaacgacgtttggtcagttccatcaacatcat agccagatgcccagagattagagcgcatgacaagta aaggacggttgtcagcgtcataagaggtttttac |
| Oligonucleotide | ds98-931 | This paper | | Homologous sequence (931 nt): gaacggaaaacatccttcatagaaatttc acgcggcggcaagttgccatacaaaacag ggtcgccagcaatatcggtataagtcaaag caccttttagcgttaaggtactgaatctcttta gtcgcagtaggcggaaaacgaacaagcgc aagagtaaacatagtgccatgctcaggaaca aagaaacgcggcacagaatgtttataggtctg ttgaacacgaccagaaaactggcctaacgacg tttggtcagttccatcaacatcatagccagatg cccagagattagagcgcatgacaagtaaagg acggttgtcagcgtcataagaggtttttacctcc aaatgaagaaataacatcatggtaacgctgc atgaagtaatcacgttcttggtcagtatgcaa attagcataagcagcttgcagacccataatgt caatagatgtggtagaagtcgtcatttggcga gaaagctcagtctcaggaggaagcggagcag tccaaatgtttttgagatggcagcaacggaaa ccataacgagcatcatcttgattaagctcatt agggttagcctcggtacggtcaggcatccac ggcgctttaaaatagttgttatagatattcaaa taaccctgaaacaaatgcttagggatttttatt ggtatcagggttaatcgtgccaagaaaagcgg catggtcaatataaccagtagtgttaacagtcgg gagaggagtggcattaacaccatccttcatgaac ttaatccactgttcaccataaacgtgacgatgagg gacataaaaagtaaaaatgtctacagtagagtc aatagcaaggccacgacgcaatggagaaagac ggagagcgccaacggcgtccatctcgaaggagt cgccagcgataaccggagtagttgaaatggtaataagac |

*Continued on next page*

*Continued*

| Reagent type (species) or resource | Designation | Source or reference | Identifiers | Additional information |
|---|---|---|---|---|
| Oligonucleotide | ds98-915 | This paper | | Homologous sequence (915 nt): gaagtcatgattgaatcgcgagtggtcg gcagattgcgataaacggtcacattaaa tttaacctgactattccactgcaacaact gaacggactggaaacactggtcataat catggtggcgaataagtacgcgttcttg caaatcaccagaaggcggttcctgaat gaatgggaagccttcaagaaggtgata agcaggagaaacatacgaaggcgcat aacgataccactgaccctcagcaatctt aaacttcttagacgaatcaccagaacg gaaaacatccttcatagaaatttcacgc ggcggcaagttgccatacaaaacagg gtcgccagcaatatcggtataagtcaaa gcacctttagcgttaaggtactgaatctc tttagtcgcagtaggcggaaaacgaac aagcgcaagagtaaacatagtgccatg ctcaggaacaaagaaacgcggcacaga atgtttataggtctgttgaacacgaccaga aaactggcctaacgacgtttggtcagttcc atcaacatcatagccagatgcccagagatt agagcgcatgacaagtaaaggacggttgt cagcgtcataagaggttttacctccaaatg aagaaataacatcatggtaacgctgcatga agtaatcacgttcttggtcagtatgcaaatta gcataagcagcttgcagacccataatgtcaa tagatgtggtagaagtcgtcatttggcgaga aagctcagtctcaggaggaagcggagcagt ccaaatgttttgagatggcagcaacggaaa ccataacgagcatcatcttgattaagctcatt agggttagcctcggtacggtcaggcatccac ggcgctttaaaatagttgttatagatattca aataaccctgaaacaaatgc |
| Oligonucleotide | ds98-915-78ss | This paper | | Homologous sequence (915 nt): gaagtcatgattgaatcgcgagtggtcgg cagattgcgataaacggtcacattaaattt aacctgactattccactgcaacaactgaac ggactggaaacactggtcataatcatggtg gcgaataagtacgcgttcttgcaaatcacc agaaggcggttcctgaatgaatgggaagc cttcaagaaggtgataagcaggagaaaca tacgaaggcgcataacgataccactgaccc tcagcaatcttaaacttcttagacgaatcac cagaacggaaaacatccttcatagaaattt cacgcggcggcaagttgccatacaaaaca gggtcgccagcaatatcggtataagtcaaa gcacctttagcgttaaggtactgaatctctt tagtcgcagtaggcggaaaacgaacaagc gcaagagtaaacatagtgccatgctcagga acaaagaaacgcggcacagaatgtttataggt ctgttgaacacgaccagaaaactggcctaac gacgtttggtcagttccatcaacatcatagcca gatgcccagagattagagcgcatgacaagtaa aggacggttgtcagcgtcataagaggttttacct ccaaatgaagaaataacatcatggtaacgctgc atgaagtaatcacgttcttggtcagtatgcaaatt agcataagcagcttgcagacccataatgtcaat agatgtggtagaagtcgtcatttggcgagaaagc tcagtctcaggaggaagcggagcagtccaaatg tttttgagatggcagcaacggaaaccataacgag catcatcttgattaagctcattagggttagcctcgg tacggtcaggcatccacggcgctttaaaatagttg ttatagatattcaaataaccctgaaacaaatgc |
| Commercial enzyme | *Bsa*1 | New England Biolabs | Catalog: #R0535S | To linearize pBSphix1200 |

*Continued on next page*

*Continued*

| Reagent type (species) or resource | Designation | Source or reference | Identifiers | Additional information |
|---|---|---|---|---|
| Commercial enzyme | Phusion-U polymerase | Thermo Fischer | Catalog: #PN-F555S | |
| Commercial kit | Epitect Bisulfite kit | Qiagen | Catalog: #59104 | DMA |
| Commercial kit | SMRTbell Template Prep Kit 1.0 | Pacific Biosciences | Catalog: #100-259-100 | DMA |
| Commercial reagent | Sera-Mag Speed Bead Carboxylate-Modified Magnetic particles (Hydrophobic) | Sigma | Catalog: #PN-65152105050250 | DMA |
| Commercial reagent | AMPure PB | Pacific Biosciences | Catalog: #100-265-900 | DMA |
| Commercial reagent | SYBR Gold Nucleic Acid Stain | Invitrogen | Catalog: #S11494 | |

## In vitro D-loop assay ssDNA substrate preparation

ds98-*607* and ds98-*197*-78ss were created as described in *Wright and Heyer, 2014*. Additionally, ds98-*931,* ds98-*915,* and ds98-*915*-78ss substrates were created similarly. The pBSbase phagemid vector was cloned to comprise either the 931 nt or the 915 nt homologies. The 931 or 915 nt homology inserts were amplified from pBSphiX1200 by using pBS-51-931nt-subs-F and pBS-1013nt-subs-R or pBS-1013nt-subs-F and pBS-915nt-subs-R primers respectively. The substrates were then spliced out from the cloned phagemids as previously described (*Wright and Heyer, 2014*).

## In vitro D-loop formation

D-loop reactions were performed as described in *Wright and Heyer, 2014* with the following modifications. The reactions used a linearized or supercoiled pBSphix1200 plasmid donors. All D-loop reactions were carried out at 30°C. Homologous ssDNA was added at 3 nM with the 3 kb donor dsDNA also present at 3 nM (molecules). Rad51 was saturating with respect to the invading ssDNA (1 Rad51 to 3 nts ssDNA), and RPA was added at one heterotrimer to 25 nt ssDNA, while Rad54 was added at 18 nM monomers. 1x buffer was used as described in *Wright and Heyer, 2014*. The order of addition was: Rad51 + ssDNA, 10 min incubation; then RPA, 10 min incubation; and finally, Rad54 + linear dsDNA, 15 min incubation. In case a supercoiled dsDNA was used instead of a linear dsDNA donor, the reaction was carried on for 10 min to achieve maximum D-loop formation and prevent D-loop disruption based on *Wright and Heyer, 2014*. The reactions had a final volume of 25 µl. Reactions were stopped with 2 mg/ml Proteinase K (2.5 µl of 20 mg/ml Proteinase K) and 10 mM EDTA (0.5 µl of 0.5 M EDTA). The reaction was then split, such that 9 µl of the reaction was added to a tube containing 0.2% SDS (0.2 µl of 10% SDS) and 1x DNA loading dye for gel visualization (2.8 µl of 6x dye). This fraction of the reaction was deproteinized by incubating at room temperature (RT) for 1–2 hr. Subsequently, the samples were separated on a 0.8% TBE agarose gel at 70 V for ~3 hr. The gel was stained with SYBR Gold Stain for 30 min at RT, before visualization. The remainder of the D-loop reaction (19 µl) was incubated at room temperature (RT) for 30 min to allow deproteinization, before proceeding to bisulfite treatment for the D-loop Mapping Assay (DMA). Note that SDS was avoided in this fraction of the D-loop sample to prevent branch migration of D-loops (*Allers and Lichten, 2000*). The reaction volume ensures that >50 ng of dsDNA was incorporated into the bisulfite reaction.

## D-loop mapping assay (DMA)
### Non-denaturing bisulfite treatment

Once deproteinized, the D-loops were treated with 85 µl of sodium bisulfite and 35 µl of DNA protecting reagent from the Qiagen 'Epitect Bisulfite Kit' at RT for 3 hr. RT was used for the treatment instead of the recommended higher temperatures to avoid D-loop disruption and migration as well as to avoid DNA denaturation. The bisulfite reaction was stopped by adding 310 µl BL buffer spiked with 3.1 µl of carrier RNA, following the kit instructions. The RNA was added since the dsDNA

concentration is <100 ng preventing non-specific binding to the column. The bisulfite reaction was completed using the kit and finally eluted in 18 µl of EB buffer.

## Two-step PCR

For the first round of PCR, primers were designed complementary to the non-homologous regions of the dsDNA donor, more than 500 nt upstream and downstream of the homology. The primers contained a 'universal' primer sequence at their 5′-end (UNI+Donor-PB-F, UNI+Phix-PB-R). In case of the spiked in pBSKS- ssDNA used to test the bisulfite conversion efficiency (see below), the primers were modified to be unbiased, lacking a cytosine in the forward primer or guanine in the reverse primer. Thus, both the bisulfite-modified and unmodified DNA were amplified. Reactions were performed using ThermoFischer PhusionU DNA polymerase (ThermoFisher PN F555S) to produce high-fidelity, long-range single-band products, despite the presence of uracil in the bisulfite-treated DNA. 0.5 µl of eluted bisulfite DNA was used as input. PCR conditions: 98℃ 2 min, [98℃ 30 s, 55℃ 30 s, 72℃ 3 min]x35 cycles, 72℃ 10 min, store at 10℃. Reactions were purified by 0.7x Sera-Mag SpeedBead Carboxylate-Modified Magnetic particles (Hydrophobic) (PN 65152105050250) and eluted in 50 µl of 1x TE pH 7.5. For the second round of PCR, the primers contained the universal primer sequence and a symmetric barcode sequence at the 5′-end. Barcode sequences were obtained from https://github.com/PacificBiosciences/Bioinformatics-Training/blob/master/barcoding/pacbio_384_barcodes.fasta. 0.5 µl of PCR-1 elution was used as input. PCR conditions: 98℃ 2 min, [98℃ 30 s, 64℃ 30 s, 72℃ 3 min] x 35 cycles, 72℃ 10 min, store at 10℃. All PCR reactions were purified using 0.7x Sera-Mag beads and eluted in 50 µl of 1x TE pH 7.5.

## AMPure bead purification

AMPure PB beads from PacBio (PN 100-265-900) were used. 0.7x volume, or as specified, of AMPure beads, were added to the DNA. The DNA was allowed to bind to the beads by shaking on a VWR vortex mixer at speed-2 at room temperature. After quick centrifugation, the tube was placed on a magnetic bead rack. 30–60 s was given to allow bead separation towards the magnet, and then the supernatant was carefully removed. The beads were then washed with freshly prepared 70% ethanol. Ethanol was added without disturbing the beads, incubated for 30 s, and pipetted out. The wash was repeated. Finally, the tube was briefly centrifuged and placed on the magnetic rack to remove any residual ethanol. After a 30–60 s air-dry, the beads were resuspended in 1x TE pH 7.5. The beads were shaken on the vortex for 2–3 min to allow elution of DNA, before placing them back on the rack to pipette out the DNA. DNA concentration was measured by NanoDrop.

## Amplicon pooling

In order to multiplex the sequencing, PCR amplicons from various D-loop reactions were pooled before library preparation. Barcoded amplicons that were AMPure purified from the second PCR reaction were pooled in equimolar concentration. Ideally, for a Sequel-I or Sequel-II run, the amplicons were pooled to have a final concentration of 2–3 µg DNA. We usually pooled approximately 40–60 samples (depending on the PacBio system used for sequencing, see below). The pooled amplicons were concentrated using a 0.7x AMPure bead wash and eluted in 37 µl of EB buffer (obtained from SMRTbell Template Prep Kit 1.0). The DNA concentration was measured by Nanodrop to have at least 2 µg DNA.

## SMRTbell library preparation

The PacBio Sequel or RS-II system was used to achieve long-read, single-molecule resolution sequencing of D-loop footprints. SMRTbell Template Prep Kit 1.0 (PN 100-259-100) from PacBio was used for preparing the libraries. Libraries were built as per the 'Procedure and Checklist – 2 kb Template Preparation and Sequencing' protocol (PN 001-143-835-08) from PacBio with some modifications. The DNA repair reaction was carried out for 30 min at 37℃. Ligation was done for 1 hr instead of 15 min at 25℃. Finally, the DNA library was eluted in 12 µl of Epitect Bisulfite buffer. SMRTbell libraries were quantified by Qubit or NanoDrop, and their size was cross-validated using gel electrophoresis or Agilent Genomic's 2100 Bioanalyzer. Libraries were sequenced on a PacBio RS-II, Sequel-I or -II instrument with 10 hr movie times and a V2 primer.

Note that depending on the PacBio sequencing instrument used, the quality and quantity of output varied. For RS-II runs, only 8–10 samples were multiplexed and generated 500–1,000 reads per sample. For Sequel-I runs, ~50 different samples were multiplexed, resulting in ~3,000–5,000 reads per sample. For Sequel-II runs, ~50 samples were pooled as well but resulted in >20,000 reads per sample. The Sequel-II instrument may permit multiplexing ~200 samples without loss of data or undersampling. Moreover, the D-loop levels detected by the Sequel-II system were relatively lower than those quantified from Sequel-I. This might be due to oversequencing the uninvaded DNA molecules.

## Computational data processing

### Circular consensus sequence (CCS) generation

Default parameters of SMRT link v6.0 with ccs 3.0.0. commit 1035 f6f for (PacBio RS-II and Sequel-I) or v8.0 (for Sequel-II) were used for processing subreads into CCS reads. As part of default parameters, a minimum subread quality of at least 90% was processed into CCS with a minimum pass filter of 3.

### Duplicate read removal

We used dedupe2.sh from package BBMap V37.90 (https://sourceforge.net/projects/bbmap/) with default parameters except for mid = 98, nam = 4, k = 31, and e = 30. An average of <5% reads were removed as duplicates in the combined datasets.

## Gargamel computational pipeline

The Gargamel pipeline (available at https://github.com/srhartono/footLoop_sh) allows users to map reads, assign strands, call single-molecule D-loop footprints as peaks of C-to-Tconversion, perform clustering on peaks, and visualize the data, similar to the pipeline described in *Malig et al., 2020*.

### Read mapping

Reads were mapped to the dsDNA reference sequence using Bismark v0.20.0 (*Krueger and Andrews, 2011*) (part of the footLoop.pl pipeline). The reference sequence included a 100 bp buffer off the beginning and end positions. Bismark default settings were used except for a slightly relaxed minimum score threshold (`-rdg` 2,1 `-rfg` 2,1 `-score_min` L,0,–0.8) and bowtie 2.2.6 (`-bowtie2`) was used. Truncated reads shorter than 50% of their expected length were discarded. After removing truncated reads, the median length of reads was 2,500 bp, which is the expected size. Altogether, the stringent requirements imposed for circular consensus, duplicate removal, mapping, and size, ensured the selection of very high-quality reads.

### Strand assignment

For each read, strandedness was assigned based on conversion patterns. Reads with insufficient conversions (C-to-T<6 and G-to-A<6) could not be assigned and were named 'unknown'. Likewise, if the number of C-to-T conversions was within + / - 10% of G-to-A conversions on a read, then the strandedness was considered unknown. Such 'unknown' strand reads represented less than ~5% of the total pool.

Otherwise, reads with predominant C-to-Tor G-to-A conversions were assigned as non-template (top) or template (bottom) strand, respectively. Ambiguous regions carrying indels due to PacBio sequencing errors were masked (including a 5 bp buffer around the indel) so as not to distort the conversion frequency calculation. bottom-strand reads were more frequently observed (>2 fold) than the top strand (or the displaced strand of a D-loop). This is likely due to a PCR bias in amplification of uracil containing DNA and/or nicking of the displaced ssDNA on a D-loop, which is often associated with the bisulfite treatment.

### Peak calling

A threshold-based sliding window method was used to call tracts of C-to-Tconversion referred to as D-loop peak. Unless otherwise specified, the windows spanned 50 consecutive cytosines and were moved across each read one cytosine at a time. (To mitigate sequence biases, we defined the window size based on a number of consecutive cytosines rather than a fixed length of nucleotides.) For

each window, we calculated the C-to-Tconversion frequency and called a window D-loop positive if a minimum of 40% cytosines were converted. Positive windows were further extended if neighboring windows also satisfied the 40% threshold. Upon encountering a window with conversion frequency less than the threshold, the peak was terminated, and its boundaries recorded.

We tested a combination of window sizes (20, 30, and 50 cytosines) and conversion thresholds (25%, 30%, and 40%); the results were qualitatively similar except that more positive peaks were recovered for less stringent conditions. A window size of 50 cytosines and minimal C-to-Tconversion of 40% permitted a good combination of specificity and sensitivity.

### Clustering and data visualization

All D-loop peaks were clustered based on their location from 5′ to 3′ end. Reads from each strand that contained a D-loop peak were visualized as a separate footprint map. The footprint map depicts position on cytosine in reference sequence as yellow vertical lines. Each horizontal line in the footprint map depicts a different read molecule. The color of cytosine for each read is changed to green for every C-to-T modification. For a stretch of C-to-T conversions that cross the peak threshold, the color is changed to red, representing a footprint. A separate footprint map is formed for reads that contain a footprint, and for reads devoid of a footprint. Reads representing the top and bottom-strand are also separately visualized.

## Optimizing peak threshold and determining bisulfite conversion efficiency

We spiked in ssDNA (pBlueScript KS- circular DNA) in an in vitro D-loop sample to test bisulfite conversion efficiency and to optimize the peak calling threshold. pBSKS- ssDNA was prepared as described in *Wright and Heyer, 2014*. The D-loop sample formed from ds98-*931* and a linear donor was spiked with 0.03 nM (1/100th of ds98-*931*/dsDNA) pBSKS- ssDNA molecules. The ssDNA was spiked along with the ds98-*931* substrate in the D-loop reaction. UNI+pBSKS-F and UNI+pBSKS-R primers (devoid of cytosines/guanines) were used to amplify the spiked ssDNA during the PCR-1 step of DMA.

## Analysis of D-loop footprints

### Quantitation of D-loops from reads

D-loops were quantified by calculating the percentage of reads from each strand (top or bottom) containing a D-loop footprint.

### D-loop length analysis

D-loop length was calculated from each D-loop footprint by subtracting the location in nucleotides of the 5′-end of the peak from the 3′-end of the peak as defined by the peak calling process. Refer to source code for the R script. The average D-loop length was calculated by averaging all the D-loop lengths from a replicate.

### Distribution of D-loop position

The distribution of D-loop position within a given sample was analyzed using the following R packages: GenomicRanges (*Lawrence et al., 2013*), readr, and tidyverse. GenomicRanges from Bioconductor was used to bin D-loops based on their position into 100 bp bins. The coverage of D-loops within each bin was calculated and normalized to the total number of D-loops for that sample. The distribution of D-loops within the homology window is depicted as a line graph. Refer to source code for the R script.

## Data availability

The computational pipeline for mapping reads, strand assignment, peak calling, clustering, data visualization, and D-loop analysis is available from GitHub (https://github.com/srhartono/footLoop_sh).

## Acknowledgements

We thank members of the Heyer laboratory, especially Aurele Piazza and William Wright, for stimulating discussions. We thank Diedre Reitz for helpful feedback on the manuscript. We also thank members of the Chedin laboratory, in particular, Lionel Sanz, for providing AMPure beads and Maika Malig for technical help. We also thank the DNA core technologies at UC Davis and the UC Berkeley Genomics Facility for providing PacBio sequencing services. This research used core services supported by P30 CA93373 and was supported by NIH grants GM58015 and CA92276 to W.-D.H. and NIH grant GM120607 to FC.

## Additional information

### Competing interests

Wolf-Dietrich Heyer: Reviewing editor, *eLife*. The other authors declare that no competing interests exist.

### Funding

| Funder | Grant reference number | Author |
| --- | --- | --- |
| National Institutes of Health | GM 58015 | Wolf-Dietrich Heyer |
| National Institutes of Health | CA 92276 | Wolf-Dietrich Heyer |
| National Institutes of Health | GM 120607 | Frédéric Chédin |
| National Institutes of Health | P30 CA 93373 | Wolf-Dietrich Heyer |

The funders had no role in study design, data collection and interpretation, or the decision to submit the work for publication.

### Author contributions

Shanaya Shital Shah, Conceptualization, Software, Formal analysis, Investigation, Visualization, Writing - original draft, Writing - review and editing; Stella R Hartono, Software, Writing - review and editing; Frédéric Chédin, Supervision, Writing - review and editing; Wolf-Dietrich Heyer, Conceptualization, Formal analysis, Supervision, Funding acquisition, Project administration, Writing - review and editing

### Author ORCIDs

Shanaya Shital Shah (iD) https://orcid.org/0000-0002-2881-2794
Wolf-Dietrich Heyer (iD) https://orcid.org/0000-0002-7774-1953

### Decision letter and Author response

Decision letter https://doi.org/10.7554/eLife.59111.sa1
Author response https://doi.org/10.7554/eLife.59111.sa2

## Additional files

### Supplementary files

• Source code 1. RScript for the D-loop Length Analysis.

• Source code 2. RScript for the analysis of D-loop position and distribution.

• Source data 1. Source data for all figures.

• Supplementary file 1. Total reads with human proteins. Table summarizing the total number of reads containing a footprint as 'peak' and the total number of reads analyzed as 'total' for each strand. % Peak' indicate the percentage of reads containing a footprint. The data represents a cumulation from two independent replicates.

• Transparent reporting form

## Data availability

All data generated or analyzed during this study are included in the manuscript and supporting files. Source data files have been provided for all numerical data.

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
