## [Decision Letter]

**Acceptance summary:**

Displacement loops (D-loops) are important intermediates in homologous recombination, translesion DNA synthesis and repair of broken DNA replication forks. Currently available methods to detect the size of the displaced strand in a D-loop have low resolution. This paper describes development of a novel in vitro technique for characterization of the length and positions of individual D-loops with near base-pair resolution, and also allows the assessment of the distribution of D-loops in a population of molecules. The displaced strand is marked by converting cytosines to uracils and mapping the D-loops through single-molecule real time sequencing. The technique will have wide application in biochemical investigations of D-loop biology, and aid in the analysis of various factors that have been proposed to act on D-loop formation, stability, and processing.

**Decision letter after peer review:**

Thank you for submitting your article "Bisulfite treatment and single-moleculereal real-time sequencing reveals D-loop length, position and distribution" for consideration by *eLife*. Your article has been reviewed by three peer reviewers, and the evaluation has been overseen by Maria Spies as the Reviewing Editor and Jessica Tyler as the Senior Editor. The following individuals involved in review of your submission have agreed to reveal their identity: Anna Malkova (Reviewer #1); Andrew J Deans (Reviewer #2).

The reviewers have discussed the reviews with one another and the Reviewing Editor has drafted this decision to help you prepare a revised submission.

Summary:

D-loops are important intermediate in homologous recombination, translesion DNA synthesis and repair of broken replication forks. Currently available methods to detect D-loops and the size of the displaced strand generally have low resolution. The reviewers and the reviewing editor agreed that the authors have developed a novel in vitro technique for characterization of the length and position of individual D-loops with near base-pair resolution, which also allows the assessment of the distribution of D-loops in a population of molecules. The displaced strand is marked by converting cytosines to uracils and mapping the D-loops by through single-molecule real time sequencing. The technique will have wide application in biochemical investigations of D-loop biology, and aid in the analysis of various factors that have been proposed to act on D-loop formation, stability, and processing.

Essential revisions:

The reviewers concur that overall, the paper will be of high interest for researchers in DNA repair and recombination. While a series of excellent controls are used to confirm the quality of this approach and no additional experiments are necessary, the reviewers felt that the authors need to better acknowledge some of the limitations of the assay. Please address the following questions:

1) The authors believe that the modifications introduced by bisulfite map only affect the final D-loop positions. Is it possible that some changes in the D-loop length, positioning, etc., can occur after the addition of bisulfite, and therefore the resulting picture represents the map of all changes that happened with D-loop during that time?

2) Please expand the explanation why a significant difference in the length distribution was observed between the ds98-*915* and ds98-*931* substrates (Figure 4E).

Similarly, subsection “Characterization of D-loops formed on a linear donor reveal enrichment of D-loops at the 3′-end and depict a wide range of D-loop lengths across the region of homology”: Please discuss the different amounts of D-loop formation for the different substrates (Figure 4B). E.g. why is D-loop formation more efficient (20.7% peaks) with the ds98-*607* substrate than with the ds98-*915* substrate (8.8% peaks)?

3) It is not clear why for human proteins the percentage of D-loops observed on a gel correlated with the frequency determined by DMA, while it was not the case for yeast proteins.

4) Figure 5—figure supplement 1: how are the "top" and "bottom" strands distinguished in this figure?

5) Figure 2A: Why the first several reads extend beyond the region of homology (to the right)?

Also, subsection “Length of D-loops with a supercoiled donor is restricted by the supercoiling density of the donor”: The authors might want to comment on the average D-loop length for the ds-98-*197*-78ss substrate being longer than the homology. Is that due to the peak calling algorithm including non-converted Cs at the window borders into the peak? If so, the authors could consider letting the peaks terminate at the last converted C in the window.

6) Figure 2—figure supplement 1D: How was the percentage of D-loops longer than 360nt calculated?

7) Figure 4—figure supplement 2B: The frequency of D-loops protruding to the right of the region of homology appears especially high. Is there a reason for this high level?

8) Subsection “Position of D-loops formed by a supercoiled donor is restricted within the homology window, while their distribution reveals an enrichment at the 3′-end of homology”: Please clarify the argument how enrichment of D-loops close to the 3'end of ssDNA supports the model that Rad51 filament grows preferentially in a 5'-3' direction. Isn't the entire ssDNA coated with RAD51 filament in this assay? Could the result point to an increased likelihood that the 3'-end is flexible or available for pairing.

---

## [Author Response]

Essential revisions:The reviewers concur that overall, the paper will be of high interest for researchers in DNA repair and recombination. While a series of excellent controls are used to confirm the quality of this approach and no additional experiments are necessary, the reviewers felt that the authors need to better acknowledge some of the limitations of the assay. Please address the following questions:1) The authors believe that the modifications introduced by bisulfite map only affect the final D-loop positions. Is it possible that some changes in the D-loop length, positioning, etc., can occur after the addition of bisulfite, and therefore the resulting picture represents the map of all changes that happened with D-loop during that time?

We agree that it would be more accurate to say that the footprints represent the map of all conversions that occurred on a D-loop over the bisulfite incubation time. However, D-loops are presumed to be relatively stable at room temperature, as no D-loop dissolution is observed over time (Wright et al., 2014). Moreover, since most of the footprints from a ds98-*931* or ds98-*607* substrate are ~200 nt in length, D-loop migration on at least a supercoiled donor appears to be minimal. Nevertheless, since this is still a possibility, especially for linear donors, we will change the statements accordingly in the manuscript and describe the caveat. See subsection “Effect of Different Peak Thresholds on D-Loop Length and Distribution”, subsection “Advantages and Limitations of the D-loop Mapping Assay” (Point 5) and subsection “D-loop Mapping Assay (DMA)”.

2) Please expand the explanation why a significant difference in the length distribution was observed between the ds98-915 and ds98-931 substrates (Figure 4E).Similarly, subsection “Characterization of D-loops formed on a linear donor reveal enrichment of D-loops at the 3′-end and depict a wide range of D-loop lengths across the region of homology”: Please discuss the different amounts of D-loop formation for the different substrates (Figure 4B). E.g. why is D-loop formation more efficient (20.7% peaks) with the ds98-607 substrate than with the ds98-915 substrate (8.8% peaks)?

D-loops formed from ds98-*915* and ds98-*931* substrates differ in mean D-loop length by 60 nt. Apart from the 16 nt difference in homology length, ds98-*915* and ds98-*931* are substrates with different sequences (with some overlapping sequence) and variable D-loop formation efficiencies. Differences in the sequence and the distribution/frequency of cytosines may impact the length of individual peaks defined. This may have a subsequent effect on the average D-loop length. For instance, ds98-*915* has within the homology sequence, 192 cytosines. While ds98-*931* has only 187 cytosines despite the longer homology. Thus, the slightly less frequent cytosine distribution in ds98-*931* may contribute to the 60 nt increase in mean D-loop length. We incorporate this explanation in the subsection “Characterization of D-loops formed on a linear donor reveal enrichment of D-loops at the 3′-end and depict a wide range of D-loop lengths across the region of homology”.

In Figure 4B: The ds98-*931* substrate forms D-loops with an efficiency comparable to the ds98-*607* substrate. This trend is captured by both gel-based assay and the DMA assay. The ds98-*915* possesses different homologous sequence in comparison to ds98-*931*, which is the likely cause for the different D-loop formation efficiency. Additionally, ds98-*915* D-loops were mapped using Sequel-II, instead of Sequel-I (used for most other D-loop samples). Sequel-II generates 5x more reads per sample than Sequel-I. Sequel-II is observed to have some bias towards sequencing the uninvaded DNA, resulting in a relatively lower quantification of the D-loop level. Similarly, two of the three independent replicates for the ds98-*915-*78ss substrate was sequenced by Sequel-II system. The D-loop level observed by the replicates from Sequel-II were lower (7%, 3.7%) than the one from Sequel-I (22.6%). Nevertheless, the trend of relative D-loop levels remains comparable to the gel-based assay. We added this caveat of Sequel systems in the Materials and methods section.

3) It is not clear why for human proteins the percentage of D-loops observed on a gel correlated with the frequency determined by DMA, while it was not the case for yeast proteins.

The percentage of D-loops observed on gel correlate with that determined by DMA with both yeast and human proteins. The D-loops from the yeast protein show a statistically significant linear correlation, with an R square value of 0.7. The correlation with human proteins visually appears stronger due to the very low D-loop formation efficiency (1.5-3%). We are currently unable to get a correlation R square value for human proteins due to low sample number (only 4 samples in total, compared to >20 samples with yeast protein).

4) Figure 5—figure supplement 1: how are the "top" and "bottom" strands distinguished in this figure?

The footprints maps depicted in Figure 5—figure supplement 1 are representative of the top strands. Thank you for pointing this out, we made the correction in Figure 5—figure supplement 1, and added description in the figure legend.

5) Figure 2A: Why the first several reads extend beyond the region of homology (to the right)?Also, subsection “Length of D-loops with a supercoiled donor is restricted by the supercoiling density of the donor”: The authors might want to comment on the average D-loop length for the ds-98-197-78ss substrate being longer than the homology. Is that due to the peak calling algorithm including non-converted Cs at the window borders into the peak? If so, the authors could consider letting the peaks terminate at the last converted C in the window.

The first several reads extend beyond the region of homology, potentially due to DNA breathing close to the 3’-end of a D-loop. If a C outside the 3’-end of homology is converted, and if the threshold permits, that C may be included in the peak defined. We clarify this now in the figure legend.

The average D-loop length for ds98-*197*-78ss is slightly longer (206 nt) than the expected 197 nt, mainly because the length of homology is very close to the threshold limit of t40w50. Within the 197 nt homology, there exists 43 cytosines, slightly lower than the minimum requirement of 50 consecutive cytosines. So, if the conversion rate is higher than 40% within these cytosines, the pipeline would stretch the D-loop length to beyond 197 nt, to incorporate the surrounding additional 7 cytosines, until the footprint crosses the threshold of t40w50. In fact, when t40w40 threshold is used, the average D-loop length is 191 ± 41 nt. We clarify this in the subsection “Length of D-loops with a supercoiled donor is restricted by the supercoiling density of the donor”.

6) Figure 2—figure supplement 1D: How was the percentage of D-loops longer than 360nt calculated?

Percentage of D-loops longer than 350 nt was calculated as the percentage of total D-loops analyzed for that sample. We clarify this now in the figure legend of Figure 2—figure supplement 1D.

7) Figure 4—figure supplement 2B: The frequency of D-loops protruding to the right of the region of homology appears especially high. Is there a reason for this high level?

There is no specific reason for it, except that the region is relatively C-rich. Although it is rare for a C-rich DNA region to undergo breathing, when breathing does occur, it very quickly leads to modification of nearby Cs due to the presence of a mismatch bubble. The modified C-rich region thus ends up crossing the threshold of t40w50 in some cases and is captured as a footprint in DMA. However, this is very rare, detected as ~60 (0.4%) footprints out of 14,790 total reads analyzed for that sample. Supporting the possibility that it arises due to DNA breathing, the bottom-strand reads also contain footprint in that region at a frequency of 0.2%. We added a representation of footprints detected on the bottom-strand for the sample in Figure 4—figure supplement 2B. We clarify this now in the figure legend, as well as in the subsection “Characterization of D-loops formed on a linear donor reveal enrichment of D-loops at the 3′-end and depict a wide range of D-loop lengths across the region of homology”.

8) Subsection “Position of D-loops formed by a supercoiled donor is restricted within the homology window, while their distribution reveals an enrichment at the 3′-end of homology”: Please clarify the argument how enrichment of D-loops close to the 3'end of ssDNA supports the model that Rad51 filament grows preferentially in a 5'-3' direction. Isn't the entire ssDNA coated with RAD51 filament in this assay? Could the result point to an increased likelihood that the 3'-end is flexible or available for pairing.

At the single-molecule level, it is possible that potentially not every single ssDNA molecule is saturated with Rad51. However, the 5’-to-3’ directional Rad51 filament growth would ensure that a Rad51 protein randomly bound on ssDNA, would result in subsequent Rad51 cooperativity covering preferentially the 3’-end of ssDNA. The relatively higher presence of Rad51 at 3’-end may enrich for D-loops in that region.

However, we agree, that it is also possible that the higher flexibility of ssDNA at the 3’-end may also increase likelihood of D-loop in that region. We added this point to the subsection “Position of D-loops formed by a supercoiled donor is restricted within the homology window, while their distribution reveals an enrichment at the 3′-end of homology”.